# Impact of pyruvic acid photolysis on acetaldehyde and peroxy radical formation in the boreal forest: Theoretical calculations and model results.

Philipp G. Eger[1], Luc Vereecken[2], Rolf Sander[1], Jan Schuladen[1], Nicolas Sobanski[1], Horst Fischer[1], Einar Karu[1], Jonathan Williams[1], Ville Vakkari[3,4], Tuukka Petäjä[5], Jos Lelieveld[1], Andrea Pozzer[1] and John N. Crowley[1]

[1]Atmospheric Chemistry Department, Max-Planck-Institute for Chemistry, 55128-Mainz, Germany
[2] Institute for Energy and Climate Research: IEK-8, Forschungszentrum Juelich, 52425 Juelich, Germany
[3]Atmospheric Composition Unit, Finnish Meteorological Institute, 00101 Helsinki, Finland
[4]Atmospheric Chemistry Research Group, Chemical Resource Beneficiation, North-West University, Potchefstroom, South Africa
[5]Institute for Atmospheric and Earth System Research (INAR) / Physics, Faculty of Science, University of Helsinki, Finland

*Correspondence to*: John N. Crowley (john.crowley@mpic.de)

**Abstract.**

Based on the first measurements of gas-phase pyruvic acid ($CH_3C(O)C(O)OH$) in the boreal forest, we derive effective emission rates of pyruvic acid and compare them with monoterpene emission rates over the diel cycle. Using a data-constrained box-model, we determine the impact of pyruvic acid photolysis on the formation of acetaldehyde ($CH_3CHO$) and the peroxy radicals $CH_3C(O)O_2$ and $HO_2$ during an autumn campaign (IBAIRN, 2016) in the boreal forest.

The results are dependent on the quantum yield ($\phi$) and mechanism of the photodissociation of pyruvic acid and the fate of a likely major product, methylhydroxy carbene ($CH_3COH$). With the box-model, we investigate two different scenarios in which we follow the present IUPAC recommendations with $\phi = 0.2$ (at 1 bar of air) and the main photolysis products (60 %) are acetaldehyde + $CO_2$ with 35% C-C bond fission to form HOCO and $CH_3CO$ (scenario A). In the second scenario (B), the formation of vibrationally hot $CH_3COH$ (and $CO_2$) represents the main dissociation pathway at longer wavelengths (~75%) with a ~25% contribution from C-C bond fission to form HOCO and $CH_3CO$ (at shorter wavelengths). In scenario 2 we vary $\phi$ between 0.2 and 1 and, based on the results of our theoretical calculations, allow the thermalised $CH_3COH$ to react with $O_2$ (forming peroxy radicals) and to undergo acid-catalysed isomerisation to $CH_3CHO$.

When constraining the pyruvic acid to measured mixing ratios and independent of the model scenario, we find that the photolysis of pyruvic acid is the dominant source of $CH_3CHO$ during the IBAIRN campaign with a contribution between ~70 and 90 % to the total production rate. We find that the photolysis of pyruvic acid is also a major source of the acetylperoxy radical, with contributions varying between ~20 and 60 % dependent on the choice of $\phi$ and the products formed. $HO_2$ production rates are also enhanced, mainly via the formation of $CH_3O_2$. The elevated production rates of $CH_3C(O)O_2$ and $HO_2$ and concentration of $CH_3CHO$ result in significant increases in the modelled mixing ratios of $CH_3C(O)OOH$, $CH_3OOH$, HCHO and $H_2O_2$.

## 1 Introduction

Organic acids play a crucial role in tropospheric chemistry, impacting secondary organic aerosol formation, air quality and climate (Kanakidou et al., 2005; Hallquist et al., 2009). Pyruvic acid ($CH_3C(O)C(O)OH$), an organic acid that is central in plant metabolism as part of the Krebs cycle (Walker, 1962), is found in tropospheric air in the gas phase as well as in the aerosol phase, especially in the boundary layer over vegetated regions. Gas-phase mixing ratios ranging from a few to several hundred parts per trillion (pptv) have been reported in various locations around the world, including the tropical rain forest, boreal forest, rural areas with temperate forest, and regions influenced by urban outflow. A recent overview of existing measurements of gas-phase pyruvic acid is given by Eger et al. (2020).

A known source of pyruvic acid is the photo-oxidation of isoprene, via the ozonolysis of methyl vinyl ketone and subsequent hydrolysis of the Criegee intermediates (Jacob and Wofsy, 1988; Grosjean et al., 1993; Paulot et al., 2009). Further potential sources are the photolysis of methylglyoxal (Raber and Moortgat, 1995), the gas-phase photo-oxidation of aromatics in the presence of $NO_x$ (Grosjean, 1984; Praplan et al., 2014), the aqueous-phase oxidation of methylglyoxal (Stefan and Bolton, 1999) and reactions taking place within biomass burning plumes (Andreae et al., 1987; Helas et al., 1992). In addition, pyruvic acid has been reported to be directly emitted from vegetation (Talbot et al., 1990; Jardine et al., 2010a; Jardine et al., 2010b; Eger et al., 2020). Compared to acetic acid, the presence of a second (non-acidic) carbonyl group imparts on pyruvic acid an absorption spectrum that extends from ultraviolet to visible wavelengths (see Fig. 1) and photolysis is a major sink of pyruvic acid in the boundary layer, with deposition and heterogeneous uptake to the aerosol phase also contributing to its removal. Photolysis of pyruvic acid in air results in a number of different radical and stable products, the major ones are expected to be acetaldehyde, $HO_2$ and $CH_3C(O)O_2$ (more details are presented in Sect. 1.1). These products can have a significant impact on tropospheric chemistry, e.g. via the formation of peroxyacetyl nitrate (PAN), peracetic acid (PAA) and formaldehyde (HCHO). Global models have recently revealed discrepancies between simulated and measured acetaldehyde concentrations (Millet et al., 2010; Wang et al., 2019; Wang et al., 2020). Wang et al. (2020) reported $CH_3CHO$ mixing ratios that were up to a factor of 10 higher than predicted by a global chemistry-transport model (EMAC) in the marine boundary layer around the Arabian Peninsula, implying missing sources of $CH_3CHO$ in remote and polluted regions. Wang et al. (2019) also found that models systematically underestimate $CH_3CHO$ compared to observations implying a missing source of acetaldehyde in the remote troposphere. This finding was supported by the simultaneous measurement of PAA (which is formed e.g. via the degradation of acetaldehyde in remote environments) with the organic aerosol source of $CH_3CHO$ also insufficient to explain the results. Instead, Wang et al. (2019) suggested that $CH_3CHO$ arises from the degradation of gas-phase organic compounds. Pyruvic acid, among other organic acids in the gas and aerosol phase, might be one of the compounds that can be converted to acetaldehyde to the remote troposphere and its integration into global models might contribute to resolve discrepancies, especially in forested regions.

Generally, field measurements as well as modelling and laboratory-based kinetic studies on pyruvic acid are limited and its impact on atmospheric chemistry is still poorly understood. In this study we highlight the potential role of pyruvic acid in the boreal forest, one of the largest terrestrial biomes on Earth. For this, we use data from a measurement campaign in 2016 (IBAIRN, Influence of Biosphere–Atmosphere Interactions on the Reactive Nitrogen budget).

## 1.1 The photolysis of pyruvic acid: Quantum yields and products

Because its reaction with OH is slow ($k_{OH+pyruvic\ acid}$ = $1.2 \times 10^{-13}$ cm$^3$ molecule$^{-1}$ s$^{-1}$ at 298 K, (Mellouki and Mu, 2003), photolysis and dry deposition are the dominant loss terms for gas-phase pyruvic acid. Heterogeneous uptake to atmospheric aerosols is also calculated to be inefficient during the IBAIRN campaign in the boreal forest (see below), where particle surface area densities were of the order of $2 \times 10^{-7}$ cm$^2$ cm$^{-3}$ and the particles contained a large organic fraction (Liebmann et al., 2019) that is likely to reduce the uptake coefficient compared to that measured for pure aqueous particles ($\gamma = 0.06$, Eugene et al. (2018)).

The photodissociation of pyruvic acid at actinic wavelengths is not well understood. According to the most recent IUPAC evaluation (IUPAC, 2020), which considers experimental data until 2017, the three thermodyamically accessible photolysis channels are:

$$CH_3C(O)C(O)OH + h\nu \quad \rightarrow \quad CH_3CHO + CO_2 \tag{R1}$$
$$\rightarrow \quad CH_3C(O)OH + CO \tag{R2}$$
$$\rightarrow \quad CH_3CO + HOCO \tag{R3}$$

IUPAC recommend a photodissociation quantum yield ($\phi$) of 0.2 at 1 bar pressure (i.e. for boundary layer conditions) with branching ratios of 0.6, 0.05 and 0.35 for reactions R1, R2 and R3, respectively, which implicitly assumes that the initially formed carbene (CH$_3$COH) immediately isomerises to acetaldehyde. The radical products of reaction R3 (CH$_3$CO and HOCO) react rapidly in air to form peroxy radicals (R4, R5).

$$CH_3CO + O_2 + M \quad \rightarrow \quad CH_3C(O)O_2 + M \tag{R4}$$
$$HOCO + O_2 \quad \rightarrow \quad HO_2 + CO_2 \tag{R5}$$

The formation of methylhydroxy carbene (CH$_3$COH) as an intermediate in pyruvic acid photolysis has been postulated for many years (Vesley and Leermakers, 1964; Yamamoto and Back, 1985). Schreiner et al. (2011), observed isomerisation of singlet CH$_3$COH to acetaldehyde in an Ar matrix at 11 K; their high level theoretical analysis revealing high barriers for isomerisation, where H-atom tunnelling though the energy barrier led to a lifetime of about 1 hour at 11 K, favouring the formation of acetaldehyde over that of vinyl alcohol. Only very recently has CH$_3$COH been detected experimentally as a product of pyruvic acid photolysis in the gas-phase (Samanta et al., 2021) and its unimolecular isomerisation to both CH$_3$CHO and CH$_2$=CHOH confirmed to be efficient at the experimental pressure of a few mbar of helium. Indeed, Samanta et al. (2021) show that, at a photolysis wavelength of 351 nm (close to the maximum cross-section of pyruvic acid) formation of an energy-

rich carbene (CH$_3$COH$^\#$) and CO$_2$ (R6) is essentially the only product channel operating. CH$_3$CHO and CH$_2$=CHOH were formed subsequently (at ~ 2:1 ratio favouring CH$_3$CHO) in the unimolecular isomerisation of CH$_3$COH$^\#$ (R7, R8).

$\quad\quad\quad$ CH$_3$C(O)C(O)OH + $h\nu$ $\quad\rightarrow\quad$ CH$_3$COH$^\#$ + CO$_2$ $\hfill$ (R6)

$\quad\quad\quad$ CH$_3$COH$^\#$ $\quad\quad\quad\quad\rightarrow\quad$ CH$_3$CHO $\hfill$ (R7)

$\quad\quad\quad\quad\quad\quad\quad\quad\quad\quad\rightarrow\quad$ CH$_2$=CHOH $\hfill$ (R8)

Samanta et al. (2021) suggest that, at ambient pressures, a significant fraction of the energised, nascent carbene will be deactivated by collisions with air (R9) and the thermalised carbene (CH$_3$COH), which can no longer rapidly overcome the

barriers to isomerisation, may react with oxygen or water vapour (Reed Harris et al., 2016; Reed Harris et al., 2017b; Eger et al., 2020; Samanta et al., 2021) (R10, R11).

$\quad\quad\quad$ CH$_3$COH$^\#$ + M $\quad\quad\rightarrow\quad$ CH$_3$COH + M $\hfill$ (R9)

$\quad\quad\quad$ CH$_3$COH + O$_2$ $\quad\quad\rightarrow\quad$ CH$_3$C(O) + HO$_2$ $\hfill$ (R10)

$\quad\quad\quad$ CH$_3$COH + H$_2$O $\quad\quad\rightarrow\quad$ CH$_3$CH(OH)$_2$ $\hfill$ (R11)

As summarised by IUPAC (2020), there have been many experimental studies deriving primary photodissociation quantum yields and product yields following the photolysis of pyruvic acid. The studies which were carried out at atmospherically relevant wavelengths (i.e. within the ~300 - 400 nm absorption band) are listed in Table S1. The experiments were carried out at different pressures of various bath gases and at different wavelengths and at different concentrations of pyruvic acid, all of which appear to play a role in determining the products formed. Table S1 shows that CO$_2$ is formed at a yield of close to 100%

whereas the yield of CH$_3$CHO is highly variable. CH$_2$=CHOH has been detected both at a few Torr of Helium (Samanta et al., 2021) and at 1bar of air (Calvert et al., 2011). Other end-products observed during the photolysis of pyruvic acid in 1 bar of air include acetic acid (Calvert et al., 2011; Reed Harris et al., 2016; Reed Harris et al., 2017b) and PAN (Grosjean, 1983; Berges and Warneck, 1992) when NO$_2$ was present, which together provide evidence for the formation of the acetyl peroxy radical (CH$_3$C(O)O$_2$), and thus CH$_3$CO, e.g., in reaction R3 and R4 when sunlight or solar-simulating light sources are used.

Further secondary products (resulting e.g. from the further reactions of CH$_3$CHO) such as HCHO and CH$_3$OH have also been observed at pressures close to 1 bar (Grosjean, 1983; Calvert et al., 2011; Reed Harris et al., 2016; Reed Harris et al., 2017a). While the Norrish type-1 process (C-C bond fission) forming CH$_3$CO and HOCO appears to be unimportant at 351 nm (Samanta et al., 2021), it may be favoured at wavelengths < 340 nm (Chang et al., 2014). This is illustrated in Fig. 1 where we present the wavelength resolved photolysis rate constants across the UV-absorption spectrum of pyruvic acid (assuming an

overall photolysis quantum yield of 1, and absorption cross-sections recommended by IUPAC). The wavelength resolved actinic flux was calculated for the IBAIRN measurement site on the 16.09.2016 using the Tropospheric Ultraviolet and Visible Radiation model (TUV, https://www.acom.ucar.edu/Models/TUV/Interactive_TUV/). Integration of the J-values at wavelengths < 340 nm, indicate that (at local noon) ≈ 25% of pyruvic acid dissociation occurs at these shorter wavelengths.

## 2 Methods

The goal of this study is to evaluate the impact of pyruvic acid on acetaldehyde and radical formation rates in the boreal forest by using a data-constrained, chemical box-model. For this purpose we make use of experimental data from a field study, which was performed in the Finish boreal forest at the "Station for Measuring Ecosystem-Atmosphere Relations II" (SMEAR II) in Hyytiälä (61.846 °N, 24.295 °E, 180 m above sea level, see Hari and Kulmala (2005)), an area that is characterised by large emission rates of biogenics (mainly monoterpenes) and low $NO_x$ concentrations (Rinne et al., 2000; Williams et al., 2011;

Aalto et al., 2015; Fischer et al., 2021).

The variability in the reported photodissociation quantum yield and product distributions (IUPAC, 2021) suggests that pyruvic acid photodissociation is not yet fully understood, In addition, the fate of the potentially dominant carbene product (Samanta et al., 2021) is unknown. Therefore, in order to better constrain the fate of $CH_3COH$ in the atmosphere, quantum chemical calculations were undertaken to characterise its likely atmospheric reactions, for which experimental data does not exist.

**2.1. The IBAIRN campaign**

The IBAIRN campaign took place in September 2016, during the summer–autumn transition, and was characterised by frequent temperature inversions near ground level during night-time (Liebmann et al., 2018a), which led to the accumulation of nocturnally emitted trace gases from vegetation. A detailed description of the campaign and the instruments deployed can be found elsewhere (Liebmann et al., 2018a; Eger et al., 2020). A summary (with details of detection limit etc.) is provided in

Table S2. Briefly, pyruvic acid was measured by a chemical ionisation quadrupole mass spectrometer (Eger et al., 2020), the sum of monoterpenes (henceforth referred to as MT) was measured by a PTR-ToF-MS and single MTs were monitored by a GC-AED (Liebmann et al., 2018a). Despite some discrepancies related to instrument location and inhomogeneity in terpene emissions within the forest, both instruments were in reasonably good agreement throughout the campaign. Since a high temporal resolution is preferable for our simulation, we have used the PTR-ToF-MS dataset. NO and $NO_2$ were measured by

chemiluminescence detector and cavity ring down spectrometer (Sobanski et al., 2016; Liebmann et al., 2018b), ozone was measured by optical absorption and CO by Quantum-cascade-Laser absorption spectroscopy (Eger et al., 2020). Formic and acetic acid as well as methyl-ethyl-ketone (MEK) and methyl-vinyl-ketone (MVK) were taken from the continuous PTRMS measurements at the site at heights between 42 and 336 m. Photolysis rate coefficients were derived using actinic flux measurements from a spectral radiometer (METCON GmbH) (METCON GmbH) and evaluated cross sections and quantum

yields (Burkholder et al., 2015). Mixing layer (MXL) heights were derived by combining in-situ measurements made by a scanning Doppler lidar (Hellén et al., 2018) with results from the ECMWF ERA-Interim reanalysis (Dee et al., 2011) with a spatial resolution of ~80 km. Since the lidar was unable to resolve MXL heights < 60 m (as regularly experienced during nocturnal inversions), all values below this threshold have been set to 60 m, representing an upper limit.

**2.2 Theoretical analysis of the fate of singlet methylhydroxy carbene, $CH_3COH$**

We investigated the reactions of $CH_3COH$ theoretically under atmospheric conditions, examining its unimolecular reactions, and bimolecular reactions with $O_2$, $H_2O$ and $HC(O)OH$, where the latter is representative of carboxylic acids. The reaction with pyruvic acid itself is also briefly explored. The rovibrational characteristics and energetics of all critical points on the potential energy surface were characterized at the CCSD(T)/aug-cc-pVTZ//M06-2X-D3/aug-cc-pVTZ level of theory with a wavenumber scaling factor of 0.971 (Zhao and Truhlar, 2008; Dunning, 1989; Purvis and Bartlett, 1982; Grimme et al., 2011;

Database of Frequency Scale Factors for Electronic Model Chemistries (Version 4); Alecu et al., 2010). This method compares favourably with the more rigorous focal point analysis of Schreiner et al. (2011), with energy differences in the singlet state unimolecular chemistry of less than 0.7 kcal mol$^{-1}$, indicating that the method is reliable for kinetic predictions under atmospheric temperatures. Where necessary, broken symmetry SCF calculations were used to describe singlet biradicals (Noodleman, 1981), and IRC calculations were used to verify the pathways. For reactants, products and transition states we

exhaustively characterized all conformers; for complexes we only characterized those directly connecting to a transition state. All quantum chemical calculations were performed using the Gaussian-16 program suite (Frisch et al., 2016).

The quantum chemical data was then used to calculate high-pressure rate coefficients for reactions over a saddle point using multi-conformer transition state theory (MC-TST) calculations (Vereecken and Peeters, 2003), under a rigid rotor harmonic oscillator approximation. Tunnelling corrections are performed assuming an asymmetric Eckart barrier (Eckart, 1930; Johnston

and Heicklen, 1962). Most reactions have high energy barriers, and the presence of pre- and post-reaction complexes has negligible influence on the reaction rate. For barrierless reactions, typically complexation reactions, we assume the reaction rate is close to the collision limit unless indicated otherwise.

**2.3 Box model**

We have used the CAABA/MECCA atmospheric chemistry box model to numerically simulate the impact of pyruvic acid

photolysis on the formation of radicals and $CH_3CHO$ over the diel cycle during the IBAIRN campaign. Our study is based on model version 4.4.2, with updated reactions related to pyruvic acid in which two different scenarios were investigated (see section 3.3) in order to examine the sensitivity of the model output to e.g. photolysis quantum yields and products.

The chemical mechanism used in this study contains >600 gas-phase species and ~2000 gas-phase reactions and photolysis steps. In addition to the basic ozone, HOx and NOx chemistry, the mechanism contains the detailed "Mainz Organic

Mechanism" (MOM) for non-methane hydrocarbons (NMHC), isoprene, terpenes, and aromatics. MOM is derived from a reduced version of the Master Chemical Mechanism (MCM). Full details about CAABA/MECCA and MOM are available in Sander et al. (2019). Photolysis reactions are calculated for a latitude of 62 °N. A complete reaction scheme and source of rate coefficients can be found in the data archive (see "data availability").

Several parameters (temperature, pressure, relative humidity), and trace-gas concentrations (pyruvic acid, $O_3$, NO, $NO_2$, PAN, CO, MTs, formic and acetic acid, methyl-ethyl-ketone (MEK), methyl-vinyl-ketone (MVK)) as well as the photolysis rate constants of various trace-gases were constrained to values measured during the IBAIRN campaign.

Based on the GS-AED measurements, the MTs were split into α-pinene (49 %), β-pinene (13 %), Δ-carene (27 %) and camphene (8 %). Limonene is not included in the standard chemical mechanism of CAABA/MECCA but as its contribution to the MTs during IBAIRN was only 3 % it was treated as Δ-carene (increasing the Δ-carene contribution to 30 %).

The atmospheric methane mixing ratio was set to a constant value of 1.8 ppmv. Non-methane alkanes, the degradation of which represents ~ 30–45 % of the acetaldehyde source globally (Millet et al., 2010) were constrained to 1000 pptv of ethane, 250 pptv of propane and 150 pptv of n-butane, as found in similar environments in Finland (Hakola et al., 2006; Hellén et al., 2015). The mixing ratio of PAN, which is generally the most abundant of peroxy acetyl nitrates (PNs), was calculated from a measurement of the sum of peroxy nitrates whereby $[PAN] = 0.9 \times \Sigma[PNs]$ (estimation based on observations by e.g. Shepson et al. (1992b), Roberts et al. (2004) and Roiger et al. (2011)). The model-generated, averaged OH concentration through the diel cycle was in good agreement (within ~20%) with that calculated from the correlation of ground-level OH measurements with UVB radiation intensity at the Hyytiälä site (with $[OH] = 5.62 \times 10^5 [UVB]^{0.62}$ molecule $cm^{-3}$ when UVB is in units of W $m^{-2}$ (Petäjä et al., 2009; Hellén et al., 2018)) but showed more variability resulting from changes in NO mixing ratios and the conversion of $HO_2$ to OH.

# 3 Results and discussion

In the following, we analyse in-situ measurements of pyruvic acid to derive emission rates, present the results of the theoretical calculations of the fate of $CH_3OH$ and discuss the box-model output for the IBAIRN campaign with a focus on pyruvic acid emission rates and its impact on acetaldehyde and radical chemistry in the boundary layer of the boreal forest.

## 3.1 Pyruvic acid emission rate relative to monoterpenes during IBAIRN

In order to derive the pyruvic acid emission rate ($E_{pyr}$) during IBAIRN we assume that only photolysis and dry deposition contribute significantly to its overall loss rate and that pyruvic acid is in steady-state. The latter assumption is reasonable as its mean lifetime was $(2 \pm 0.5)$ hours. Due to a homogeneous fetch at the measurement site we can neglect transport processes and $E_{pyr}$ is defined by Eq. (1), where $[pyr]_{ss}$ is the measured concentration, $J_{pyr}$ is the photolysis rate constant of pyruvic acid, $k_{dep}$ is the first-order loss rate constant for its dry deposition, and $h_{MXL}$ is the well-mixed boundary layer height.

$$E_{pyr} = [pyr]_{ss} \left( J_{pyr} + k_{dep} \right) h_{MXL} \tag{1}$$

$E_{pyr}$ is effectively an emission rate normalised to the MXL height ($h_{MXL}$) and has units of pptv $s^{-1}$ m. As the photolysis is a substantial fraction of the overall losses of $CH_3C(O)C(O)OH$, the choice of quantum yield φ directly impacts the calculated emission rate.

The deposition rate of pyruvic acid was calculated from $k_{dep} = v_{dep}\, h_{MXL}^{-1}$ during day and $k_{dep} = 2\, v_{dep}\, h_{MXL}^{-1}$ during night

(Shepson et al., 1992a), with the transition following the diel variation in the mixing layer height $h_{MXL}$ (see Fig. S1 in the Supplement). Further, as pyruvic acid and $H_2O_2$ have similar solubilities, we assumed that their deposition velocities are equal, so that $v_{dep} = 8.4$ cm s$^{-1}$ during day and $v_{dep} = 0.8$ cm s$^{-1}$ during night, as derived by Crowley et al. (2018) for the same site. This resulted in a minimum dry-deposition loss rate constant of $k_{dep} = 0.9 \times 10^{-4}$ s$^{-1}$ during day and a maximum of $k_{dep} = 1.8 \times 10^{-4}$ s$^{-1}$ during night.

The same calculation is performed for the MTs ($E_{MT}$) over the same period (and thus for the same MXL height). We note that $h_{MXL}$ controls not only the value of $k_{dep}$ but also directly affects the mixing ratios of both MTs and pyruvic acid for a given emission rate. The relative emission rate ($E_{pyr} / E_{MT}$) can be calculated from Eq. (2) where terms in square brackets are concentrations.

$$\frac{E_{pyr}}{E_{MT}} = \frac{[pyr]_{ss}\,(J_{pyr}+k_{dep})}{[MT]_{ss}\,(k_{OH}[OH]+k_{NO3}[NO_3]+k_{O3}[O_3])} \qquad (2)$$

In the denominator, $k_{OH}$, $k_{NO3}$ and $k_{O3}$ are rate coefficients for reaction of MTs with OH, NO$_3$ and O$_3$, respectively. As we do not have GC data at high time resolution, an effective rate coefficient for loss of the monoterpenes was derived from the mean MT composition as measured by GC-AED (49 % α-pinene, 13 % β-pinene, 27 % carene (sum of 2-carene and 3-carene), 3 % $d$-limonene and 8 % camphene) and the corresponding rate coefficients (Perring et al., 2013; Gaona-Colman et al., 2017; IUPAC, 2020). This will introduce significant uncertainty (factor ~2) into the calculation of the MT emission rates. Further

uncertainty arises from the measurement of pyruvic acid, ΣMT, OH, O$_3$, NO$_3$, $h_{MXL}$ and $J_{pyr}$. In particular, the results are very sensitive to the deposition velocity ($v_{dep}$) of pyruvic acid which is an estimate based on the deposition velocity of $H_2O_2$ which itself has an uncertainty of ~ 90 % (Fischer et al., 2019). Further, our calculations are based on the assumption that the sources of pyruvic acid and MT are evenly distributed and measurements made at ~ 8.5 m above the ground are representative of the entire boundary layer (i.e. that the boundary layer is well-mixed, including the very shallow boundary layer at night). A gradient

in pyruvic acid mixing ratios at night cannot be ruled out, which would impact on our results. We estimate that the emission ratio ($E_{pyr} / E_{MT}$) in Eq. (2) is associated with an overall uncertainty of a factor ~3 notwithstanding the use of different quantum yields (and thus J-values) for pyruvic acid photolysis.

A time series of pyruvic acid and MT mixing ratios along with the MXL height ($h_{MXL}$) derived from a lidar measurement and from the ERA-Interim re-analysis is shown in Fig. S2 of the Supplementary Information. Whereas both MXL height datasets agree very well during the night when the MXL is shallow (usually < 100 m), the lidar data is on average a factor of ~2 lower

during day and characterised by a much higher variability. For the derivation of the diel profile of $h_{MXL}$ (Fig. S1) we took an average of both datasets. The diel variation displayed in Fig. S2, with the highest MT mixing ratios at night, is characteristic for this boreal forest site and has been observed in earlier studies (Hellén et al., 2018).

In the following, we focus on the mean, diel profiles of $E_{pyr}$, $E_{MT}$, $J_{pyr}$, T and $h_{MXL}$ for the IBAIRN campaign, which are

presented in Fig. 2. A plot showing the variability of the MT and pyruvic acid mixing ratios over the same period was previously reported (see Fig. 3 of Eger et al. (2020)).

During September, the emission rate of pyruvic acid ($E_{pyr}$) reaches its maximum a few hours after solar noon when the temperature peaks, similar to $E_{MT}$. However, the amplitude of the day-to-night difference in $E_{pyr}$ is a factor of ~3 smaller than observed for $E_{MT}$. This could indicate that pyruvic acid emissions are less temperature-dependent than MT emissions (see below) and that other environmental factors might additionally play a role at this time of year.

The emission rates of the MTs derived as described above show a large day-night variation with a factor ~20 larger values around noontime compared to midnight. This is significantly larger than the expected variation (factor 2–3) based on the average noon-to-midnight temperature difference of 10 K and the parameterisation of Guenther et al. (1993) whereby $E_{MT} \propto \exp(\beta(T - 297 \text{ K}))$ with $\beta = 0.1$ K$^{-1}$ (which is in line in with the empirical value of $\beta = 0.12$ K$^{-1}$ that was derived for this site in September by Hellén et al. (2018)). One potential reason for this discrepancy may be emissions in autumn from fresh leaf litter that significantly contribute to the observed mixing ratios (Hellén et al., 2018) and that the assumption of evenly distributed sources and a well-mixed boundary layer is not necessarily valid during night, especially during strong temperature inversions. Fig. S3 in the Supplement shows that the daytime emission of pyruvic acid relative to MT ($E_{pyr}$ / $E_{MT}$) varies by a factor of ~ 2, depending on the chosen scenario, whereas the nighttime emission ratio is only dependent on the deposition velocity of pyruvic acid. For further analysis we adopt a quantum yield of 0.2 (IUPAC, 2020). On average ($E_{pyr}$ / $E_{MT}$) ~ 0.6 with a minimum value of ~ 0.3 in the evening and a maximum value of ~ 1 in the early morning, indicating elevated pyruvic acid emissions relative to MT at night. To derive a *T*-dependent expression from the diurnal profile of the emission factor, we fit an exponential function to the plot of temperature versus $E_{pyr}$ / $E_{MT}$ (Fig. S4), yielding:

$$E_{pyr} = \left[ 0.28 + 3.17 \times \exp\left(\frac{273 - T}{4.24}\right) \right] \times E_{MT} \qquad (3)$$

We note that (like the values of $E_{pyr}$) the temperature dependence derived is strongly influenced by the diel variation of the MXL height and thus carries significant uncertainty and may not be transferable to other locations or even times of the year.

As our measurements of pyruvic acid are the first to have been made in the boreal forest, we cannot compare our relative emission ratio ($E_{pyr}$ / $E_{MT}$) with previous measurements in a similar environment. Instead, where possible, we derive the emission ratio from measurements of MTs, isoprene and pyruvic acid in warmer climates.

Jardine et al. (2010b) performed measurements in an enclosed (glass dome) tropical forest biome at Biosphere 2 in Arizona, US, where they found maximum mixing ratios of 120 ppbv isoprene, 6 ppbv MTs and 15 ppbv pyruvic acid. As the glass dome absorbed actinic wavelengths and prevented active photochemistry, the chemical loss processes for pyruvic acid, isoprene, and MT (including photolysis and reactions with OH, $O_3$ and $NO_3$ ) are negligible. Initially disregarding the deposition of isoprene and MT, we derive lower limits of ($E_{pyr}$ / $E_{iso}$) ~ 0.17 and ($E_{pyr}$ / $E_{MT}$) ~ 4 (see Table 1). However, due to the presence of large concentrations of isoprene-consuming microbes in the soil of Biosphere 2, the isoprene loss rate via deposition may be enhanced, which will decrease the effective emission ratio ($E_{pyr}$ / $E_{iso}$). In addition, branch enclosure studies were performed on a *mangifera indica* (mango) tree within Biosphere 2, yielding mean fluxes (in nmol m$^{-2}$ s$^{-1}$) of 3.2 for isoprene, 0.09 for MT and 0.15 for pyruvic acid. Pyruvic acid emissions peaked during the day when temperature and photosynthetically active radiation (PAR) were highest and correlated very well with isoprene emissions and (to a certain extent) with MT emissions.

Assuming that a mango tree is representative for the tropical vegetation, we derive emission ratio of ($E_{pyr}$ / $E_{iso}$ ~ 0.05 and ($E_{pyr}$ / $E_{MT}$) ~ 1.7 (see Table 1), which is consistent with our estimations for the IBAIRN campaign. However, given that Talbot et al. (1990) observed great variability in pyruvic acid emission fluxes among five different tree species during measurements in the tropical Ducke Forest Reserve close to Manaus, Brazil, this agreement may, to some extent, be coincidental. Talbot et al. (1990) also reported a mean emission flux (derived from enclosure experiments) relative to isoprene of ($E_{pyr}$ / $E_{iso}$) ~ 0.003,

which is about one order of magnitude smaller than in the study of Jardine et al. (2010b). In a further branch enclosure study by Jardine et al. (2010a) emissions from a creosotebush (*Larrea divaricata*), which is typically found in US drylands, were investigated. Average noontime branch emission rates (in $\mu$g C gdw$^{-1}$ h$^{-1}$) of 7.5, 10.4 and 0.2 for isoprene, MT and pyruvic acid resulted in relative emission ratios of ($E_{pyr}$ / $E_{iso}$) ~ 0.05 and ($E_{pyr}$ / $E_{MT}$) ~ 0.07 for this mixed isoprene-MT-emitting species.

The comparison with the few datasets available in the literature indicates that the variability of the emission factors ($E_{pyr}$ / $E_{MT}$) and ($E_{pyr}$ / $E_{iso}$) among different plant species and different environments is large. In addition, a lack of pyruvic acid measurements over different seasons in the boreal forest means that we cannot exclude that the value we derive is biased by emissions (e.g., from ground-level, decaying plant-litter in September) that are peculiar to this season and environment. The emission rates we derive are therefore relevant for the autumnal boreal forest but require validation before being extended to

other regions and seasons with confidence.

## 3.2 Theoretical calculations on the fate of CH$_3$COH

    Singlet methylhydroxy carbene, CH$_3$COH, is best characterized as having an sp$^2$-hybridized central carbon, bearing an in-plane lone pair in an sp$^2$ orbital and an empty p-orbital perpendicular to the CCO plane. The lone pairs of the hydroxy O-atom back-donate into the empty p-orbital, such that the most favourable geometry has the hydroxy-H-atom into the CCO plane.

The orientation of the terminal OH group has a large impact on the energy, with 3 kcal mol$^{-1}$ energy difference between the *syn-* and *anti-*conformers. Due to the interaction between the hydroxy O-atom and the carbene functionality, internal rotation of the OH group has a very high barrier, 24 kcal mol$^{-1}$. Concomitantly, *syn/anti-*interconversion is very slow, with predicted rate coefficients at 300 K of less than $10^{-2}$ s$^{-1}$. Under atmospheric conditions, thermalised *syn-* and *anti-*CH$_3$COH are thus best considered as separate species, with possibly distinct chemistry. No information is available on the relative yield of these

conformers from pyruvic acid photolysis.

### 3.2.1 Unimolecular reactions of CH$_3$COH

    Both *syn-* and *anti-*CH$_3$COH can isomerise to vinyl alcohol over high barriers $\geq$ 24 kcal mol$^{-1}$ (see Fig. 3). *Anti-*CH$_3$COH has an additional pathway for isomerisation to acetaldehyde, with a barrier of 23 kcal mol$^{-1}$. Due to these high barriers, the thermal rate of isomerisation is comparatively slow, with a 300 K rate coefficient of $\leq 4 \times 10^{-4}$ s$^{-1}$ (see Table 2). As already discussed

by Schreiner et al. (2011), formation of CH$_3$CHO from *anti-*CH$_3$COH is most favourable at low temperatures owing to a

thinner energy barrier and hence faster tunnelling. At temperatures above 260 K, we find that formation of $CH_2=CHOH$ from *anti*-$CH_3COH$ becomes dominant, with a ~3.5:1 ratio of $CH_2=CHOH$ to $CH_3CHO$ at room temperature.

Given the low predicted thermal rate coefficients, it seems unlikely that the experimentally observed acetaldehyde and vinyl alcohol in pyruvic acid photolysis are formed from isomerisation of *thermalized* $CH_3COH$. The energy distribution of energised, nascent carbenes would be rather broad as the available energy upon pyruvic acid photodissociation is distributed over all fragments and their relative motion, and the isomerisation yield would then be pressure-dependent. The $CH_3CHO$ and $CH_2=CH_2OH$ isomers formed would have enough energy to undergo keto-enol tautomerisation, but given the high barrier exceeding 55 kcal mol$^{-1}$ it is more probable they will instead be stabilized by collisional energy loss.

### 3.2.2 Reaction of CH$_3$COH with O$_2$

Under atmospheric conditions, the reaction with $O_2$ is potentially an important loss process for $CH_3COH$ (Reed Harris et al., 2016; Reed Harris et al., 2017a; Eger et al., 2020). The potential energy surface is shown in Fig. 4. Contrary to radicals, which react with $O_2$ by (near-)barrierless radical recombination, the singlet $CH_3COH$ carbene does not have an unpaired electron and the reaction proceeds mostly by association of its out-of-plane empty p-orbital with a lone electron pair in $O_2$, requiring orbital rearrangement to a triplet $C^\bullet OO^\bullet$ moiety with a sp$^3$-hybridized central carbon. This unfavourable process has high barriers, > 9 kcal mol$^{-1}$, and concomitantly low rate coefficients, $k$(298 K) ~10$^{-20}$ cm$^3$ molecule$^{-1}$ s$^{-1}$ (see Table 2). The rate coefficient is however highly uncertain owing to an uncertainty (~1 to 2 kcal mol$^{-1}$) on the barrier height.

The decomposition of the $CH_3C^\bullet(OH)OO^\bullet$ triplet diradical intermediate, forming $CH_3C^\bullet=O + HO_2$, is reminiscent of the chemistry of α-OH alkyl radicals with unpaired electrons, and should occur rapidly owing to the sufficiently high energy content of the peroxyl-alkyl diradical (Hermans et al., 2005, 2004; Dillon et al., 2012; Olivella et al., 2001; Dibble, 2002). Note that this chemistry is very distinct from that of the singlet $CH_3C(OH)OO$ Criegee intermediate. The acyl radical product is expected to recombine rapidly with a second $O_2$ molecule, forming acylperoxy radicals, $CH_3C(=O)OO^\bullet$. Alternatively, the triplet $CH_3C^\bullet(OH)OO^\bullet$ intermediate can react with a second $O_2$ molecule by a barrierless recombination reaction (Fig. 4), forming the diperoxy singlet diradical $CH_3C(OH)(OO^\bullet)OO^\bullet$ which in turn can eliminate $HO_2$, similarly as other α-OH peroxy radicals, forming the acylperoxy radicals directly. This second $O_2$ addition is sufficiently exothermic to allow formation of peracetic acid with a singlet $O_2$ molecule, but this process has a rather large barrier of ~24 kcal mol$^{-1}$ and is expected to be a minor contributor, leaving $CH_3C(O)O_2 + HO_2$ as the likely dominant products of the overall reaction of $CH_3COH$ with oxygen molecules.

### 3.2.3 Reactions of CH$_3$COH with carboxylic acids

Samanta et al. (2021) observed loss of $CH_3COH$ via reaction with pyruvic acid, which may indicate that its fate in the atmosphere may also be (partially) controlled by similar reactions. To theoretically investigate the reaction of $CH_3COH$ with carboxylic acids, we used formic acid in the calculations. Not only is formic acid an abundant organic acid in the atmospheric

boundary layer, its reactivity is related to the properties of the –C(=O)OH moiety, and the results are transferable to other oxoacids, including pyruvic acid, which was present in high concentrations in most laboratory investigations.

As shown in Fig. 5, $CH_3COH$ forms strong complexes with $HC(O)OH$, with 11 kcal mol$^{-1}$ stability. From this complex an addition process occurs that is best described as the transfer of the acidic $H^+$ atom to the carbene lone pair on the $CH_3COH$ central carbon, with simultaneous association of one of the negatively charged lone electron pair of the carbonyl oxygen to the carbene vacant p-orbital, forming a 1-hydroxyethylester (CH3CH(OH)OC(O)H). Due to the concerted association of the two carbene orbitals with suitable partners in the carboxylic moiety, this process has a very low barrier ( $\leq$ 1 kcal mol$^{-1}$). This mechanism is feasible due to the size of the –C(O)OH group, and the possibility of shifting the double bond to the other oxygen atom upon H-atom loss. For the anti-$CH_3COH$ carbene, we also found that an in-plane approach of the carboxylic acid towards the COH moiety in methylhydroxy carbene can simultaneously transfer the acidic H-atom to the carbene carbon while the carbene hydroxy H-atom is transferred to the carbonyl oxygen in the acid, reforming the $HC(O)OH$ co-reactant. This catalysis reaction converts anti-$CH_3COH$ to acetaldehyde, $CH_3CHO$, without an energy barrier. Both adduct formation and the catalysis reaction should proceed with rate coefficients near the collision limit.

Carboxylic acids can also catalyse keto-enol tautomerisation, possibly helping the isomerisation between $CH_3CHO$ and $CH_2=CH_2OH$ by reducing the effective barrier by over 50 kcal mol$^{-1}$ though the thermal reaction remains slow (see Table 2). The only reaction of $CH_3COH$ that has been investigated experimentally to date is that with pyruvic acid (Samanta et al., 2021), supplemented in this work by a theoretical exploration in the supporting information. Note that the large rate coefficient for $CH_3COH$ with organic acids calculated here would imply that reaction of *thermalised* $CH_3COH$ with pyruvic acid would overwhelm any other bimolecular $CH_3COH$ reaction in their work and most of the experiments listed in Table S1.

### 3.2.4 Reactions of $CH_3COH$ with $H_2O$

Based on the reactivity of small carbenes towards closed-shell molecules, Samanta et al. (2021) suggested that reaction with $H_2O$ might be an important loss process of the $CH_3COH$ carbene intermediate. We have characterized the insertion reaction of $CH_3COH$ in the $H_2O$ molecule, and found very high barriers, $\geq$ 11 kcal mol$^{-1}$, with very low rate coefficients $\sim 10^{-20}$ cm$^3$ molecule$^{-1}$ s$^{-1}$ (see Fig. 5 and Table 2). The reaction is significantly slower than with carboxylic acid as the smaller $H_2O$ molecule is unable to simultaneously reach both carbene orbitals in a favourable geometry. The reaction of $H_2O$ with $CH_3COH$ is best described as a shift of an $H^+$ atom to the carbene lone pair orbital, followed by migration of the water $HO^-$ moiety to the vacant carbene orbital to form a bond with a lone electron pair. The reaction is further hindered by the back-donation of the $CH_3COH$ oxygen atom into the vacant carbene orbital, partially filling the vacant carbene orbital and reducing the reactivity of the carbene functionality. We therefore propose that $CH_3COH$ will be significantly less reactive towards closed shell species than the $^2CH$ and $^1CH_2$ carbenes which are known to exhibit very fast insertion and cyclo-addition reactions (Vereecken et al., 1998; Goulay et al., 2009; Douglas et al., 2019; Jasper et al., 2007; Gannon et al., 2010).

### 3.2.5 Summary of theoretical calculations: The fate of *thermalised* $CH_3COH$ in 1 bar of air

The theoretical analysis of the fate of $CH_3COH$ carbene intermediates formed in PA photolysis indicates that the acetaldehyde formation observed in many experiments could be the result of a fast catalysis reaction of $CH_3COH$ with pyruvic acid, which under typical experimental conditions exceeds competing reactions, such as with $O_2$, by several orders of magnitude. This conclusion is consistent with the experimental observations of Reed Harris et al. (2017a) who report a reduction in the acetaldehyde yield when low pyruvic acid concentrations are used and an increase in the formation of acetic acid (which can be formed in the reaction of $CH_3C(O)O_2$ radicals with $HO_2$). In the atmospheric boundary layer atmosphere, where the concentrations of organic acids may lay between $10^{10}$ and $10^{11}$ molecule $cm^{-3}$ (Millet et al., 2015) and that of $O_2$ is close to 5 $\times 10^{18}$ molecule $cm^{-3}$ the reactions of $CH_3COH$ with organic acids and $O_2$ are competitive, whereas reaction of $CH_3COH$ with water is minor. Table 2 lists the predicted rate coefficients for these reactions.

### 3.3 Box-model results: Contribution of pyruvic acid to acetaldehyde and radical formation

To account for the large variability in photodissociation quantum yields and product yields reported in the literature (see above), we modelled two scenarios A and B:

**Scenario A**: In this scenario we used pyruvic acid cross sections, quantum yields and product yields according to the IUPAC recommendations (IUPAC, 2020) with a photodissociation quantum yield ($\phi$) of 0.2 at 1 bar pressure and branching ratios of 0.6, 0.05 and 0.35 for reactions R1, R2 and R3 as listed in section 1.1.

**Scenario B:** Here we use the same absorption cross-sections as scenario A but build on the recent observations of (Samanta et al., 2021) and the theoretical work presented in section 3.2, which considers the formation and fate of an excited $CH_3COH$ molecule (+ $CO_2$). In scenario B, we consider the effects of using photodissociation quantum yields of 0.2, 0.5 and 1 (scenarios $B_{0.2}$, $B_{0.5}$ and $B_1$, respectively). Photolysis at wavelengths < 340 nm was considered to generate $CH_3CO$ + HOCO, whereas photolysis at wavelengths > 340 nm was assumed to form $CO_2$ + energy rich $CH_3COH^{\#}$ which undergoes the reactions outlined in section 1.1. Assuming a quantum yield that is independent of wavelength results in 25 % of pyruvic acid photolysis at noon taking place at wavelengths < 340 nm and 75 % at wavelengths > 340 nm. In the model, we assume that this ratio does not change (i.e. we neglect wavelength dependent variations in the relative actinic flux through the diel cycle). The values of 25% and 75 % listed above roughly correspond to the relative importance of peroxy radical formation (via R3, R4 and R5) at the shorter wavelengths compared to $CH_3COH$ + $CO_2$ formation (R6) at the longer wavelengths. Some experimental data indicates that addition of $O_2$ can reduce the $CH_3CHO$ yield in favour of formation of e.g. acetic acid. For this reason we use a rate coefficient for reaction of $CH_3COH$ with $O_2$ that is competitive with the reaction between $CH_3COH$ and organic acids. This is a factor ~10 larger than the value obtained theoretically, but we consider this value still within the uncertainty (~1 to 2 kcal $mol^{-1}$ on the barrier height) of our current theoretical results as the peculiar wavefunction of $CH_3COH$ may require even higher levels of theory to be described accurately.

In the box-model, in addition to reaction with $O_2$, the thermalised carbene also reacts with formic and acetic acids to form acetaldehyde:

$$CH_3COH + HCOOH \rightarrow CH_3CHO + HCOOH \tag{R12}$$

$$CH_3COH + CH_3C(O)OH \rightarrow CH_3CHO + CH_3C(O)OH \tag{R13}$$

with a rate coefficient of $5 \times 10^{-11}$ $cm^3$ molecule$^{-1}$ s$^{-1}$.

We assumed that (at 1 bar) 70% of $CH_3COH^{\#}$ was quenched to $CH_3COH$, 20% isomerised to $CH_3CHO$ and 10% isomerised to $CH_2=CHOH$ in order to reproduce the $CH_3CHO$-to-$CH_2=CHOH$ ratio reported by Samanta et al. (2021). A summary reaction scheme for the photodissociation of pyruvic acid and the fate of the initial products is given in the SI.

### 3.3.1 $CH_3CHO$ formation

The modelled formation of $CH_3CHO$ from pyruvic acid photolysis through the diel cycle when considering scenario A is displayed as a stacked plot of contributing reactions in Fig. 6. Immediately apparent from this figure is the dominance of pyruvic acid photolysis compared to all other processes. Under scenario A, even with the low quantum yield ($\phi = 0.2$) recommended by IUPAC, pyruvic acid photolysis contributes > 80 % to the overall $CH_3CHO$ production term, with a maximum of ~15% (at noon) arising from reactions of the ethylperoxy radical, formed in the reaction of OH with ethane (7.5%) and butane (3.75 %) and in the photolysis of $CH_3C(O)C_2H_5$ (3.75 %).

Under scenario B, pyruvic acid photolysis still dominates the formation of $CH_3CHO$, with a noon-time contribution of 91%, 86 % and 71% when quantum yields of 1, 0.5 and 0.2 are considered. Of the pyruvic acid contribution, 45% of the $CH_3CHO$ arises via isomerisation of the initially formed, energised carbene (blue), while the remaining 55 % results from reactions of the thermalised carbene with formic (orange) and acetic (green) acids, the concentrations of which were constrained by observations. The modelled, noon-time mixing ratio of $CH_3CHO$ varies from 400 pptv (scenario $B_1$) to 160 pptv (scenario $B_{0.2}$) when pyruvic acid photolysis is included and is reduced to ~100 pptv when the quantum yield is set to zero. Unfortunately, reliable measurements of the $CH_3CHO$ mixing ratios with which to compare the model simulations were not available for the IBAIRN campaign as the in-situ PTRMS data set (*m/z* 45) varied from -400 to + 400 pptv over the diel cycle. The modelled, maximum mixing ratio of $CH_3CHO$ increase from ~100 pptv when pyruvic acid photolysis is neglected to > 400 pptv under scenario $B_1$. (see Fig. S

### 3.3.2 $CH_3C(O)O_2$ formation

The $CH_3C(O)O_2$ radical is formed in a termolecular reaction between the $CH_3CO$ radical and $O_2$. Figure 7 displays the main photochemical reactions that lead to the formation of $CH_3CO$ in our model. The spikes in the simulated production rates are connected to spikes in the diel average NO mixing ratio at the site. In analysing the data we therefore consider not only the contributions at noon (when, NO mixing ratios were large) but also at 10:30 when NO mixing ratios were comparably low.

Under scenario A, where $\phi = 0.2$ and the yield of the $CH_3CO$ radical is 0.35, the contribution of pyruvic acid photolysis to the overall production rate at 12:00 and 10:30 are about 23 % and 16 %, respectively, which are roughly equally divided into a direct contribution (J43018) and an indirect contribution (G42008a) arising via enhanced $CH_3CHO$ levels. The main contributors to the formation of $CH_3CO$ are reactions initiated by the degradation of isoprene and MTs (in the legend to Fig. 5: BIACETO2, C511O2, C716O2, CO23C4CHO, CO235C6CHO) which involve reactions of peroxy radicals with NO.

Under scenario $B_1$, the photolysis of pyruvic acid become significantly more important, contributing a total of 63% of the total production rate for $CH_3CO$ at 10:30 and 42% at 12:00. When considering scenarios $B_{0.5}$ and $B_{0.2}$ the contributions of pyruvic acid photolysis are reduced to 46 % (29%) and 29% (17%), respectively, where the numbers in parentheses are for the "high NOx" situation. Generally, the reaction of the thermalised carbene with $O_2$ (G42099), the direct photolysis at wavelengths < 340 nm (J43018) and the indirect enhancement in $CH_3CO$ formation via the enhanced levels of $CH_3CHO$ (G42008a) contribute roughly equally to the formation of $CH_3CO$ resulting from pyruvic acid photolysis. The modelled mixing ratio of the $CH_3C(O)O_2$ radical at noon increases by a factor ~1.5 when comparing scenario B1 with the quantum yields for pyruvic acid photodissociation set to zero.

### 3.3.3 HO$_2$ formation

In Fig. 8 we plot the nine most important model pathways to $HO_2$ production through the diel cycle. The dominant modelled production terms for $HO_2$ involve HCHO (photolysis HCHO and reaction with OH, G4108, J41001b), the reaction of methoxy radicals (G4118, whereby $CH_3O$ is generated mainly in the reaction of $CH_3O_2$ radicals with NO) and the reaction of OH with CO. The direct contribution of pyruvic acid photolysis to $HO_2$ formation (via its photolysis (J43018) and through the reaction of $CH_3COH$ with $O_2$ (G42099)) is ~ 10% under scenario $B_1$ under low NOx conditions (i.e. at 10:30). Under all other scenarios it is lower with values (in percent) of <1 (scenario A at both 10:30 and 2:00), ~6 (scenario B1 at 12:00), ~5 and ~3.5 (scenario $B_{0.5}$ at 10:30 and 12:00, respectively) and ~1.5 and <1 (scenario B0.2 at 10:30 and 12:00, respectively). However, although the direct impact of pyruvic acid photolysis is weak, it has a significant indirect effect via the enhanced formation of $CH_3C(O)O_2$ radicals (directly via R3 + R4 and R10 and indirectly via $CH_3CHO$ formation) which, in the presence of $O_2$ reacts with NO to form $CH_3O_2$. Enhanced production rates of $CH_3O_2$ result in enhanced production rates of $CH_3O$ and HCHO and thus $HO_2$.

| | | | | |
|---|---|---|---|---|
| $CH_3C(O)O_2 + NO (+O_2)$ | $\rightarrow$ | $CH_3O_2 + NO_2 + CO_2$ | | (R14) |
| $CH_3O_2 + NO$ | $\rightarrow$ | $CH_3O + NO_2$ | | (R15) |
| $CH_3O + O_2$ | $\rightarrow$ | $HCHO + HO_2$ | | (R16) |
| $HCHO + h\nu (+ 2 O_2)$ | $\rightarrow$ | $2 HO_2 + CO$ | | (R17) |

The model simulations have shown that the photolysis of pyruvic acid at the levels observed during the IBAIRN campaign have a potentially significant effect on both $CH_3CHO$ mixing ratios and production rates of $HO_2$ and $CH_3C(O)O_2$ radicals, the latter being especially enhanced under low-$NO_X$ conditions. The enhanced production rates and concentrations of $CH_3C(O)O_2$ and $HO_2$ also results in significant increases in the modelled mixing ratios of several trace gases that are formed from these radicals. When comparing scenario $B_1$ to the case in which the pyruvic acid photodissociation quantum yield ($\phi$) is set to zero

results in an increase by factors of 2.2, 2.0 and 1.6 for $CH_3C(O)OOH$, $CH_3OOH$ and $H_2O_2$, respectively (see Fig S6). HCHO mixing ratios are enhanced by a factor 1.2. Vinyl alcohol mixing ratios of up to 40 pptv were generated in scenario B1. Clearly, the photolysis of pyruvic acid can potentially impact strongly on the concentrations of e.g. C1 and C2 carbonyl compounds and peroxides in the boreal environment.

## 4 Conclusions

We have combined measurements of pyruvic acid in an autumn campaign in the boreal forest (IBAIRN) with theoretical calculations designed to characterise the fate of the methylhydroxy carbene radical ($CH_3COH$, the major product of pyruvic acid photodissociation) with a box modelling study. We investigated the impact of pyruvic acid photolysis on the rates of production of acetaldehyde ($CH_3CHO$) and the peroxy radicals $CH_3C(O)O_2$ and $HO_2$. The theoretical study revealed unexpected features of $CH_3COH$ chemistry, including slow reactions of thermalised carbene with $H_2O$ but an efficient acid-

catalysed conversion to $CH_3CHO$ in the presence of organic acids such as $HC(O)OH$. The reaction of $CH_3COH$ with $O_2$ is slow, but will contribute to its fate (and thus the formation of $CH_3C(O)O_2$ and $HO_2$) in the lower atmosphere where $O_2$ concentrations are high if the rate constant used (elevated by an order of magnitude compared to the theoretical value) is correct In our box-model, the photolysis of pyruvic acid was parameterised as presently recommended by IUPAC (whereby the main products are $CH_3CHO$ and $CO_2$) and also using a more detailed mechanism in which the formation and fate of $CH_3COH$ was

considered and in which the quantum yield was varied. In all scenarios, we find that the photolysis of pyruvic acid was the dominant source of $CH_3CHO$ during IBAIRN and that its instantaneous contribution to the daytime formation of $CH_3C(O)O_2$ varied between 16 and 63 %, dependent on the assumed scenario and also on the NO concentration. Pyruvic acid photodissociation results in a significant increase in the mixing ratios of several carbonyl compounds and peroxides in the boreal environment.

The results of our modelling study are strongly dependent on the chosen quantum yields and photodissociation mechanism. To reduce the uncertainty in the role of pyruvic acid photolysis, there is an urgent need for further experimental and theoretical work on the photochemistry of pyruvic acid and on the fate of methylhydroxy carbene under atmospheric conditions. In addition, further measurements of pyruvic acid mixing ratios and of its deposition velocity in different environments are required to better constrain its abundance, lifetime and thus the impact of its photolysis. Enclosure studies would be helpful to

investigate the dependence of pyruvic acid emission rates on different plant types and environmental conditions.

## Data availability

The Max Planck Institute data used for the IBAIRN analysis and the reaction scheme used in the box-model are archived at https://doi.org/10.5281/zenodo.3254828 (Crowley and Fischer, 2019).

## Author contributions

PE was responsible for the pyruvic acid measurement during IBAIRN. PE and JC, with contributions from JL wrote the manuscript. LV made the theoretical calculation on the fate of methylhydroxy carbene, RS and AP did the box-modelling, NS was responsible for the CRDS measurements of $NO_2$ and PANs during IBAIRN. JS was responsible for the $O_3$ and J-value measurements during IBAIRN. HF was responsible for the NO and CO measurements during IBAIRN. EK and JW were responsible for the MT measurements during IBAIRN. VV was responsible for the mixing layer height measurements during 510 IBAIRN. TP was responsible for the SMEAR II observations and infrastructure. All authors contributed to the paper.

## Competing interests

The authors declare that they have no conflict of interest.

## Acknowledgements

We thank the technical staff of SMEAR II station for the excellent support during IBAIRN.

## Financial support

We are grateful to ENVRIplus for partial financial support of the IBAIRN campaign.

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

**Table 1: Emission rate of pyruvic acid ($E_{pyr}$) relative to isoprene ($E_{iso}$) and MT ($E_{MT}$)**

| Reference | Location | Plant species | ($E_{pyr}$ / $E_{iso}$) | ($E_{pyr}$ / $E_{MT}$) |
|---|---|---|---|---|
| This study | Hyytiälä, Finland | Boreal forest | ~ 20 | 0.62 |
| Talbot et al. (1990) | Manaus, Brazil | Tropical forest | 0.003 | - |
| Jardine et al. (2010b) | Biosphere 2, Arizona, US | Tropical biome | 0.17 | 4 |
| Jardine et al. (2010b) | Biosphere 2, Arizona, US | Mango tree | 0.05 | 1.7 |
| Jardine et al. (2010a) | Biosphere 2, Arizona, US | Creosotebush | 0.05 | 0.07 |

695

**Table 2. Theory-predicted high-pressure rate coefficients for reaction of singlet CH₃COH**

| Reactants | Products | $k$(298 K) | $A$ | $n$ | $E_a$ |
|---|---|---|---|---|---|
| syn-CH₃COH + O₂ | CH₃C(OH)OO• | $2.2 \times 10^{-20}$ | 5.74E-22 | 3.05 | 4092 |
| anti-CH₃COH + O₂ | CH₃C(OH)OO• | $6.6 \times 10^{-21}$ | 1.71E-22 | 2.97 | 3960 |
| syn-CH₃COH + H₂O | CH₃CH(OH)₂ | $1.9 \times 10^{-20}$ | 1.57E-55 | 13.56 | -1049 |
| anti-CH₃COH + H₂O | CH₃CH(OH)₂ | $5.7 \times 10^{-21}$ | 1.09E-61 | 15.61 | -1443 |
| syn-CH₃COH | anti-CH₃COH | $8.9 \times 10^{-3}$ | 7.86E-20 | 10.77 | 6598 |
| | CH₂=CHOH | $1.9 \times 10^{-4}$ | 3.62E-91 | 34.20 | -1444 |
| anti-CH₃COH | syn-CH₃COH | $2.8 \times 10^{-5}$ | 6.55E-20 | 10.71 | 8137 |
| | CH₂=CHOH | $9.2 \times 10^{-5}$ | 2.02E-114 | 40.40 | -6660 |
| | CH₃C(=O)H | $3.4 \times 10^{-4}$ | 1.26E-81 | 30.96 | -563 |
| CH₂=CHOH + HCOOH | CH₃C(=O)H + HCOOH | $2.9 \times 10^{-18}$ | 1.82E-76 | 19.88 | -6192 |
| CH₃C(=O)H + HCOOH | CH₂=CHOH + HCOOH | $8.1 \times 10^{-27}$ | 1.09E-78 | 20.59 | -633 |

Calculations were performed at the CCSD(T)//M06-2X-D3 with MC-TST level of theory. Rate coefficient are given at 298 K ($s^{-1}$ or $cm^3$ molecule$^{-1}$ s$^{-1}$). Temperature dependent rate coefficients can be calculated using the parameters of a Kooij expression $k$(200-450 K) = $A \times (T/K)^n \times \exp(-E_a/T)$ with $A$ in $s^{-1}$ or $cm^3$ molecule$^{-1}$ s$^{-1}$ and $E_a$ in K.

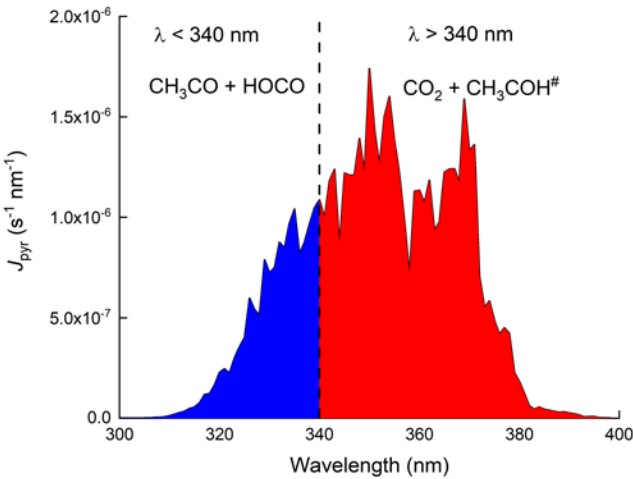

**Figure 1:** Wavelength resolved photolysis rates ($J_{pyr}$) for 13.09.2016 at solar noon. $J_{pyr}$ was calculated using a photolysis quantum yield of 1 and the absorption cross sections at 298 K preferred by IUPAC (2020).

715

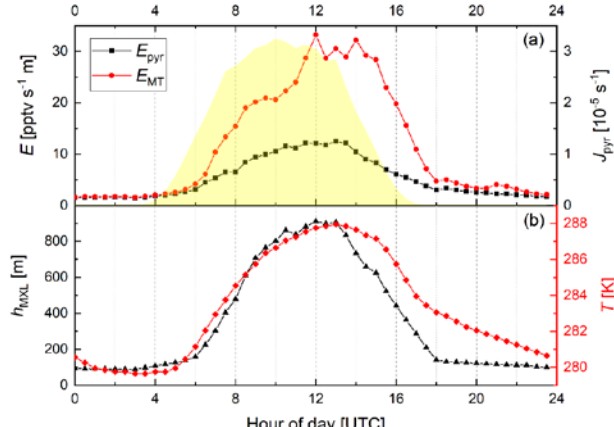

**Figure 2:** Diel variation of the (MXL height-corrected) emission rates of pyruvic acid ($E_{pyr}$, scenario B) and monoterpenes ($E_{MT}$) along with $J_{pyr}$ (yellow shaded), $T$ and $h_{MXL}$ for the IBAIRN campaign.

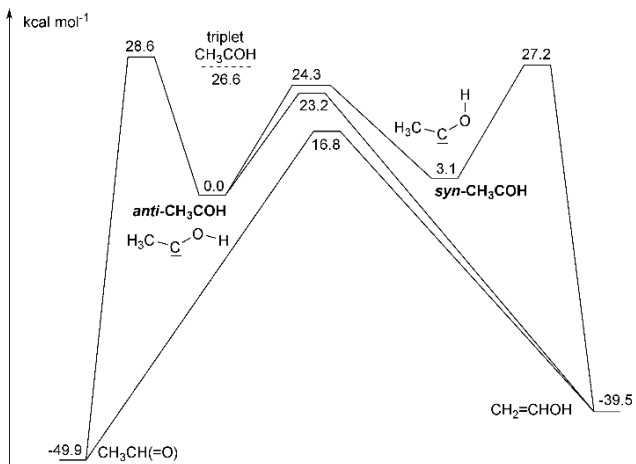

**Figure 3:** ZPE-corrected potential energy surface for unimolecular reactions of singlet
$CH_3COH$ at the CCSD(T)//M06-2X-D3 level of theory.

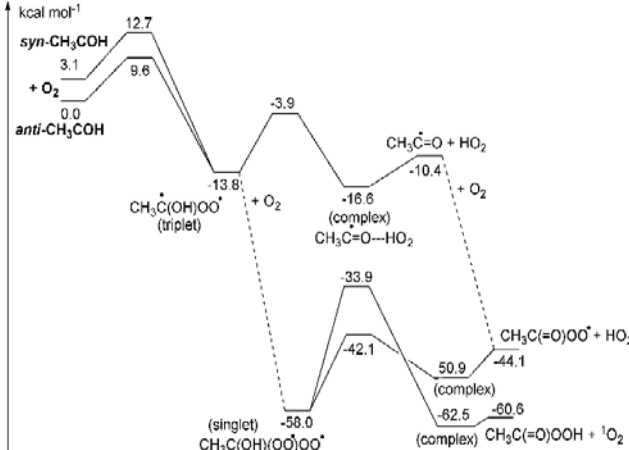

**Figure 4**: ZPE-corrected potential energy surface for reaction of singlet
CH₃COH with O₂ at the CCSD(T)//M06-2X-D3 level of theory.

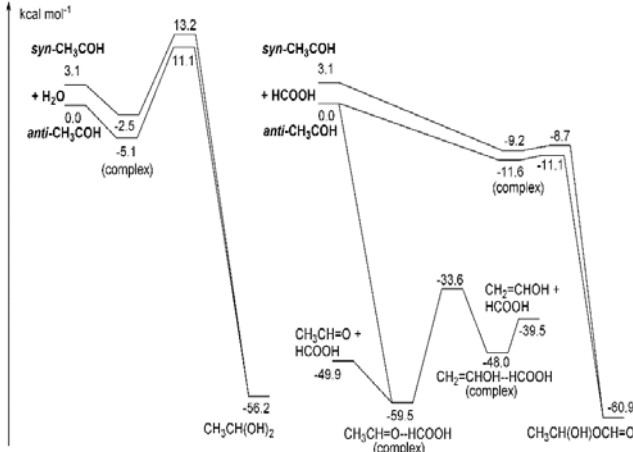

**Figure 5**: ZPE-corrected potential energy surface for reactions of singlet CH₃COH with H₂O (left) and HCOOH (right) at the CCSD(T)//M06-2X-D3 level of theory.

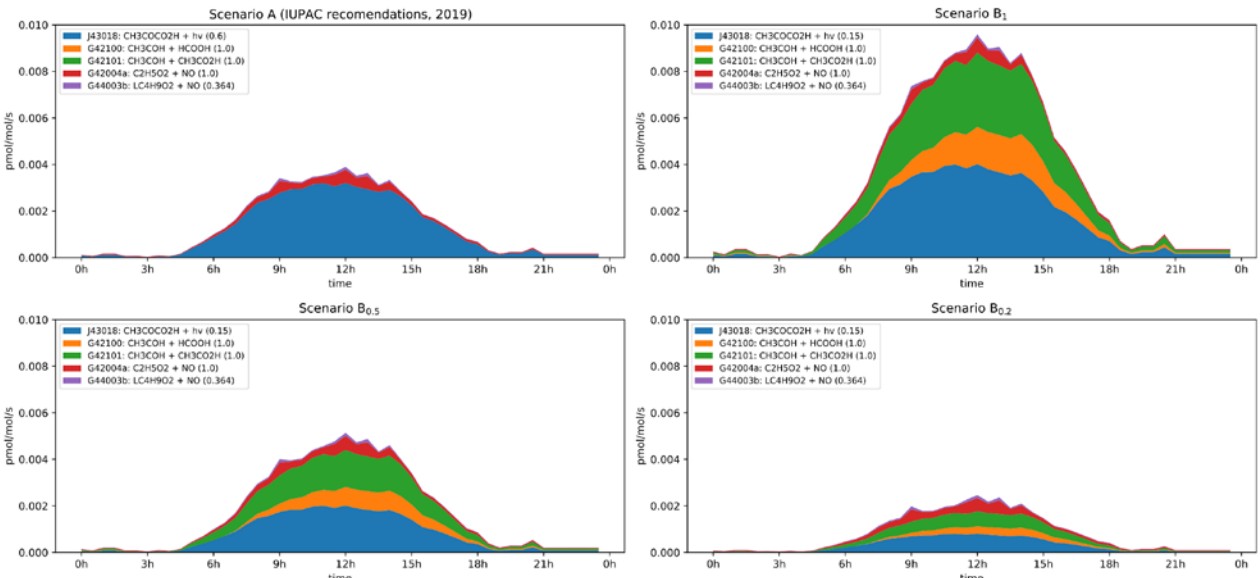

**Figure 6**: Modelled rates of CH₃CHO formation (in ppt per seconds) through the diel cycle from photolysis of pyruvic acid (blue, orange and green) and other reactions during IBAIRN. Top left: Scenario A (IUPAC recomendations from 2019). Top right: Scenario B₁. Bottom left: Scenario B₀.₅. Bottom right: Scenario B₀.₂. In the legend, the first term is the equation tag used by CAABA/MECCA for the reaction. LC4H9O2 is the peroxy radical formed in the reaction of OH with butane. A full listing of the reactions can be downloaded (see "data availability").

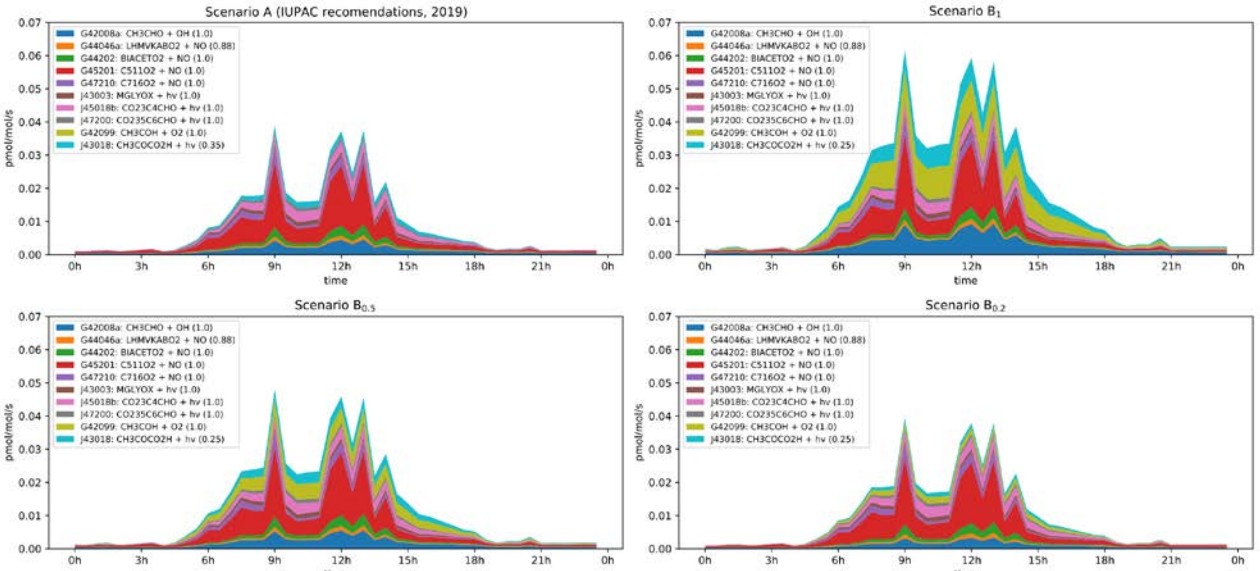

**Figure 7**: Modelled rates of CH₃CO formation (in ppt per seconds) through the diel cycle from photolysis of pyruvic acid (blue, orange and green) and other photochemical processes during IBAIRN. Top left: Scenario A (IUPAC recomendations from 2019). Top right: Scenario B₁. Bottom left: Scenario B₀.₅. Bottom right: Scenario B₀.₂. In the legend, the first term is the equation tag used by CAABA/MECCA for the reaction. A full listing of the reactions can be downloaded (see "data availability").

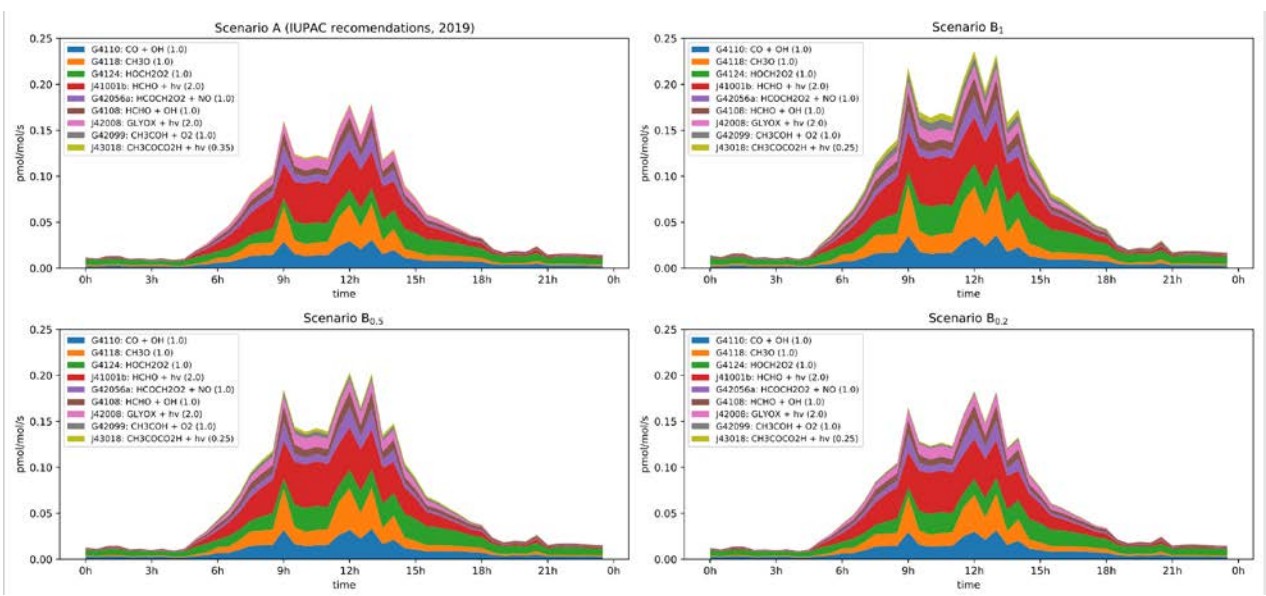

**Figure 8**: Modelled rates of $HO_2$ formation (in ppt per seconds) through the diel cycle from photolysis of pyruvic acid (blue, orange and green) and other photochemical processes during IBAIRN. Top left: Scenario A (IUPAC recomendations from 2019). Top right: Scenario $B_1$. Bottom left: Scenario $B_{0.5}$. Bottom right: Scenario $B_{0.2}$. In the legend, the first term is the MCM designation for the reaction. A full listing of the reactions can be downloaded (see "data availability").