# Peer review of "Impact of pyruvic acid photolysis on acetaldehyde and peroxy radical formation in the boreal forest: Theoretical calculations and model results."

_Atmospheric Chemistry and Physics, 2020_

## Referee Comment (RC1) · Anonymous Referee #1 · 3 Dec 2020

General comments

Building upon the results reported in a previous study (Eger et al., doi.org/10.5194/acp-20-3697-2020), this paper presents a modeling analysis of the potential impact of pyruvic acid photolysis on the chemistry and composition of the boreal forest boundary layer, for the conditions of summer and autumn campaigns at the SMEAR II field station in Hyytiälä, Finland. An observationally-constrained box model is used to investigate the contributions of pyruvic acid photolysis to the formation of acetaldehyde and peroxy radicals ($HO_2$ and $CH_3C(O)O_2$), and these contributions are reported to be significant and potentially dominant. The analysis takes account of reported large disagreements

in the overall quantum yield and product channel contributions for pyruvic acid photolysis, and therefore highlights an urgent need for further experimental studies on the photochemistry of this species.

As with Eger et al. (2020), this paper highlights a potentially important role for pyruvic acid in the boreal forest environment. A difficulty I have is whether the reported observationally-constrained modeling study is genuinely robust enough to allow the reported quantitative conclusions to be drawn, and therefore if this paper builds substantially upon the information already reported by Eger et al. (2020). I have some serious concerns about the simplicity of the chemical mechanism used and the organic chemistry it represents. As presented (Table S2), the mechanism is a substantially simplified representation of the likely set of processes that were actually occurring in the vicinity of the campaign site. While the use of simplified chemistry can be fully adequate and justifiable, there is only limited discussion of and justification for the processes that are included and (more importantly) those that are omitted in the present work. As a result, I find it quite difficult to judge how reliable some of the reported conclusions are, particularly those that relate to radical sources and contributions (see further below).

In addition, the complete set of observations that are ideally required to allow the model to be constrained are not available for either of the autumn (IBAIRN) or summer (HUMPPA) campaigns. In particular, the abundance of pyruvic acid itself was not measured during HUMPPA, but was derived from the inferred emission rate of monoterpenes using a parameterization based on the autumn IBAIRN campaign. Eger et al. (2020) report that (unlike monoterpenes) pyruvic acid emission depends on both T and PAR, and the present paper indicates that the IBAIRN parameterization may not be transferable to other times of the year (page 8, line 3). Despite this, this is exactly what is done for the HUMPPA simulation without any further discussion, justification or caveats. It is noted that the most impressive result (i.e. 94 % of acetaldehyde formation) is derived from that analysis and appears in the Abstract. Conversely, acetaldehyde was not measured during IBAIRN, and the reported contributions of pyruvic

acid photolysis for that campaign only take account of the acetaldehyde sources that are represented in the model, which are probably incomplete.

Although the presented work, and the previous study of Eger et al. (2020), provide interesting evidence for an important role for pyruvic acid in the boreal forest environment, the model used in the present modeling study is too simplistic to allow the reported quantitative conclusions to be drawn. The authors should consider addressing the shortcomings and omissions in the chemical mechanism and represented precursor species, possibly by using a customized version of an existing tool such as the MCM. I agree with the recommendations for further experimental studies on the photochemistry of pyruvic acid and on its emission rate, but these recommendations already appear in Eger et al. (2020).

Specific comments

The chemical mechanism used for the reported simulations (Table S2) contains only selected reactions. Some of these have incomplete product sets and some are parameterized, and I can find no clear discussion or justification of what these are based on or why they are considered adequate. With the exception of pyruvic acid photolysis itself, very little is represented explicitly or fully. As a result, there is a potentially enormous amount of missing organic chemistry which could otherwise contribute to the formation of the species of interest (i.e. acetaldehyde and peroxy radicals, including HO2 and CH3C(O)O2), suggesting that the simulated contributions arising from pyruvic acid photolysis are consistently overestimated.

One very clear indication of missing organic chemistry is the CH3C(O)O2 budget reported in sections 3.2 and 3.3. Decomposition of the (observed) PAN is calculated to be the major or dominant CH3C(O)O2 source. However, because PAN is a reservoir (rather than a primary source) this is approximately balanced by CH3C(O)O2 loss by reaction with NO2, indicating that an equivalent primary source is required from elsewhere. This is clearly not fully represented in the simulations. As discussed in

the literature (e.g. Fischer et al., doi.org/10.5194/acp-14-2679-2014), the sources can include oxidation or photolysis of co-called "immediate" precursor carbonyls (e.g. acetaldehyde, methylglyoxal, acetone, MEK) and a suite of terpene and isoprene oxidation products. In practice, $CH_3C(O)O_2$ can be formed from the reactions of $O_3$ with a-pinene, limonene, 2-carene and 3-carene (i.e. the measured terpenes and also many other BVOCs), from the further chemistry of peroxy radicals containing the $C(OO)C(=O)CH_3$ substructure. These are formed as co-products with OH, following decomposition of relevant Criegee intermediates and reaction of the resultant vinoxy fragment with $O_2$. The relevant chemistry in the applied chemical mechanism,

$O_3$ + terpenes —-> OH

therefore omits the C10 organic radical co-product and all its associated organic chemistry. The missing chemistry for this pathway (and for other pathways) not only includes sources of $CH_3C(O)O_2$ and other peroxy radicals, but the set of ozonolysis pathways also potentially produces "immediate" $CH_3C(O)O_2$ precursors such as methylglyoxal (and larger $C(O)C(O)CH_3$ species) and acetone. In general terms, the OH- $O_3$- and $NO_3$-initiated chemistry represented for monoterpene oxidation is severely limited, inadequately parameterized or completely absent, and the chemistry of other BVOCs (e.g. sesquiterpenes) is also not considered, even though reported to be significant at SMEAR II (Hellén et al., doi.org/10.5194/acp-18-13839-2018).

It is noted that the authors confirm that there must be missing sources of $CH_3C(O)O_2$ in their model (final sentence of section 3.3), to account for the observed formation of $CH_3C(O)OOH$ during HUMPPA reported by Crowley et al. (doi:10.5194/acp-18-13457-2018). In view of this, it is not clear why the reported contributions of pyruvic acid photolysis to $CH_3C(O)O_2$ formation (e.g. in the Abstract) are not adjusted downwards, or at least qualified, to reflect this.

Other than pyruvic acid photolysis, the only sources of acetaldehyde represented in the model appear to be the reactions of OH with ethane, propane and n-butane. These are

highly parameterized, only making fractional yields of acetaldehyde and no other products, and incorrectly acting as a radical sink. Other than a brief footnote to Table S2, no justification for this representation is given. There are potentially other precursors to acetaldehyde that may be individually or collectively important, including ethanol, larger oxygenates (e.g. propanal and MEK: Hellén et al., doi.org/10.5194/acp-4-1771-2004), any species with a C=CHCH3 substructure (e.g. propene, 2-butenes, 2-butenal, 2-hexenal: Hellén et al., doi.org/10.5194/bg-3-167-2006, doi.org/10.5194/acp-4-1771-2004) and additional alkanes to those already represented.

Other comments

Page 2, line 13: Should probably also include CO2 for completeness. The description in section 2.2.1 also identifies formation of CH3C(O)OH + CO as an "important" channel. If this is the case, should these products also be listed here?

Page 2, line 21: CH3C(O)O2 and HO2 (i.e. the other pyruvic acid photolysis products of interest) are also more immediate precursors to PAA.

Page 3, line 2: Is "large" emphasizing that the biogenics are large (i.e. monoterpenes rather than isoprene) or indicating that the emissions are large? It is not clear.

Page 3, line 15: Presumably, photolysis rates of other species were also required and were/could be calculated in the same way. The reaction listing in Table S2 also includes photolysis of glyoxal and H2O2 (although the former photolysis rate is based on that of NO2), but somewhat surprisingly not CH3CHO. There are probably other omissions too, such that the species for which photolysis is represented seem rather arbitrary.

Page 4, lines 17-32: Some of the presented information would seem to require associated citations. Currently, there are none.

Page 4, reaction (R5): The products of this reaction should be HO2 + CO2.

Page 4, line 31: For clarity and consistency, "CH3CO3 + HO2" should be written "CH3C(O)O2 and HO2".

Page 5, line 6: "UPAC".

Page 5, line 9, and Table 1: "CH3CO3" should be "CH3C(O)O2" for consistency. Please also check whole paper for consistency.

Page 5 and Table 1: The considered products of pyruvic acid photolysis are given as either "CH3CHO" or "CH3C(O)O2 + HO2". Although these are the products of interest, they do not describe either the primary photolysis products, or the full set of products following secondary chemistry (which I think are "CH3CHO + CO2" and "CH3C(O)O2 + HO2 + CO2").

Page 5, lines 1-10. From what is written, it is not clear why the IUPAC recommendation differs so much from the recent study of Reed Harris et al. (2017). Perhaps the studies on which the IUPAC recommendation is based should also be cited and described, as IUPAC presumably judged those to be more reliable and convincing.

Pages 5 and 6, sections 2.2.2 and 2.2.3. It should probably be stated again clearly what measurements were used to constrain the model for each of the campaign simulations, so that the additions and omissions can be placed in context.

Page 5, line 24: "OH" should be "The concentration of OH".

Page 5, lines 28 and 29: The basis for the assigned additional OH reactivity for unmeasured OVOCs sounds rather arbitrary. In practice, there could an abundance of both missing sources and missing sinks of OH that are unaccounted for in the simple model used.

Page 6, line 27: delta-limonene should probably be d-limonene. Is delta-carene 2-carene or 3-carene?

Page 10, line 2: Should "preceding" be "proceeding"?

Page 11, lines 14, 16 and 22. Insert "photolysis" after "pyruvic acid".

---

## Referee Comment (RC2) · Anonymous Referee #2 · 4 Feb 2021

The authors use a box model to analyze data from two field studies in Hyytiala (summer-autumn of different years), examining the potential impact of pyruvic acid (PYR) photolysis on acetaldehyde and HOx radical budgets in this boreal forest area. The topic is of interest and importance, and thus relevant to ACP readership. I have some significant concerns with the work, however, mostly related to assumptions made in the box model analyses; some of these are discussed by the authors, but I think there are others that need to be addressed (particularly in the context of how quantitative the conclusions are, given the uncertainties in PYR photolysis and depositional loss and the nature of some of the approximations made). More details follow.

[Figure]

One over-arching comment: Some of the ideas explored here have already been touched upon in the authors' previous publication on this topic (Eger et al., ACP 2020), and I found that some of the questions that came to mind were actually addressed in the previous paper. I would thus like to see a clear distinction made between the two works, via the addition of a paragraph or two to the end of the introductory material to summarize the findings of the previous study, and to set the stage for what is done in this paper.

More detailed comments:

Is it realistic to assume a mixed boundary layer (even in daytime) for fairly short-lived vegetative emissions? Can the authors be more quantitative or descriptive of what the implications of this assumption are?

Is the chemistry in the box model sufficiently detailed to capture HOx radical budgets accurately in this complex terpene-driven system, and thus assess changes driven by PYR photolysis? In particular, could there not be multiple sources of CH3C(O)O2 radicals from the terpene chemistry that is not included in the model? Do we really know enough about terpene oxidation to rule out PYR production?

Regarding OH in the IBAIRN modeling, I did not quite follow sentences near the bottom of page 5. Am I correct that OH is modeled (not constrained), but is lowered in the model via addition of extra reactivity to match the UV/OH correlation determined in other campaigns at the site? Please clarify.

Bottom of page 6 - Do the monoterpenes (MT's) have significantly different lifetimes and, if so, can the anything be said about how averaging their loss rates might impact the analysis?

Page 7 - I would like to get some sense of the variability in the average diel profiles shown, particularly for the emission rates of PYR and MT (Figure 2). This variability then carries forward to the data shown in Figures S3 and S4, correct?
Bottom of page 7 - The data showing the T-dependence of the PYR to MT emissions ratio look quite convincing. Isn't it likely, however, that this ratio is influenced also by light levels, thus complicating the analysis of the T-dependence?

Page 9 - Is PAN nominally in steady-state (or close to it)? If so, I am not sure it makes sense to consider its decomposition to be a source of $CH_3C(O)O_2$ radicals?

Page 10/11 - I have significant misgivings about doing anything quantitative with the HUMPPA campaign data, given the absence of PYR mixing ratio data. Surely, and as stated (to an extent) by the authors, the emissions ratios ($E\_PYR$ / $E\_MT$) could be affected by changes in temperatures, light levels, vegetation, availability of plant litter, different soil moisture content, etc. etc. I would recommend nothing more than some qualitative estimates regarding the HUMPPA campaign data, rather than any sort of quantitative analysis. Also, I realize that inclusion/exclusion of the biomass burning impacted data made little difference to the findings, but is it not quite likely that there are biomass burning sources of acetaldehyde, MTs and possible PYR during these periods that are not factored into the analysis here?

Minor comments:

Page 4 - (R5) should have $CO_2$ as the product, not CO.

Page 5, line 6 – IUPAC is missing the I.

Page 6, line 14 – Can some data be shown to demonstrate the statement made here stating that changes in the PYR mixing ratio could be reproduced by the model?

Pg. 7, line 9 - There is roughly a factor of two uncertainty in PYR deposition, and there is roughly a factor of four difference in the overall PYR quantum yield used in the different sensitivity studies. Thus, would not the uncertainty in the emission ratio be larger than a factor of two?

Page 8, line 25 – I am getting different numbers for the emissions ratios than are given here (on the basis of the branch emission rates given). Please check / confirm.

Page 9, line 29 – I think you are meaning to say that the fractional contribution of the alkanes drastically decreases?

Lastly, a bit of an aside: The authors might not yet be aware, but there was a presentation at the AGU a couple of months ago that appears to show the hydroxycarbene $CH_3COH + CO_2$ as the major channel at 351 nm (Osborne and co-workers). I don't think any quantum yields were reported.

---

## Author Comment (AC1) · 17 Jun 2021

**Referee 1**

We thank the referee for detailed and helpful comments, which are repeated below in black text. Our replies are in blue. We note that both reviewers suggested that making quantitative statements regarding the impact of pyruvic acid photolysis on e.g. $CH_3CHO$ or radical production was not possible given the very concise reaction scheme used. In the light of these comments we have re-done the box-modelling using a comprehensive reaction mechanism drawn from the MCM.
We have also performed quantum chemical calculations to evaluate the fate of methylhydroxy carbene, now believed to be the major product of pyruvic acid photolysis at actinic wavelengths. The manuscript has thus been substantially rewritten.

General comments

Building upon the results reported in a previous study (Eger et al., doi.org/10.5194/acp20-3697-2020), this paper presents a modeling analysis of the potential impact of pyruvic acid photolysis on the chemistry and composition of the boreal forest boundary layer, for the conditions of summer and autumn campaigns at the SMEAR II field station in Hyytiälä, Finland. An observationally-constrained box model is used to investigate the contributions of pyruvic acid photolysis to the formation of acetaldehyde and peroxy radicals (HO2 and CH3C(O)O2), and these contributions are reported to be significant and potentially dominant. The analysis takes account of reported large disagreements in the overall quantum yield and product channel contributions for pyruvic acid photolysis, and therefore highlights an urgent need for further experimental studies on the photochemistry of this species. As with Eger et al. (2020), this paper highlights a potentially important role for pyruvic acid in the boreal forest environment. A difficulty I have is whether the reported observationally-constrained modeling study is genuinely robust enough to allow the reported quantitative conclusions to be drawn, and therefore if this paper builds substantially upon the information already reported by Eger et al. (2020). I have some serious concerns about the simplicity of the chemical mechanism used and the organic chemistry it represents. As presented (Table S2), the mechanism is a substantially simplified representation of the likely set of processes that were actually occurring in the vicinity of the campaign site. While the use of simplified chemistry can be fully adequate and justifiable, there is only limited discussion of and justification for the processes that are included and (more importantly) those that are omitted in the present work. As a result, I find it quite difficult to judge how reliable some of the reported conclusions are, particularly those that relate to radical sources and contributions (see further below).
We have now performed a more detailed box-modelling study (described in section 2.3) using the CAABA/MECCA atmospheric chemistry box model with >600 gas-phase species and ~2000 gas-phase reactions and photolysis steps.

In addition, the complete set of observations that are ideally required to allow the model to be constrained are not available for either of the autumn (IBAIRN) or summer (HUMPPA) campaigns. In particular, the abundance of pyruvic acid itself was not measured during HUMPPA, but was derived from the inferred emission rate of monoterpenes using a parameterization based on the autumn IBAIRN campaign. Eger et al. (2020) report that (unlike monoterpenes) pyruvic acid emission depends on both T and PAR, and the present paper indicates that the IBAIRN parameterization may not be transferable to other times of the year (page 8, line 3). Despite this, this is exactly what is done for the HUMPPA

simulation without any further discussion, justification or caveats. It is noted that the most impressive result (i.e. 94 % of acetaldehyde formation) is derived from that analysis and appears in the Abstract. Conversely, acetaldehyde was not measured during IBAIRN, and the reported contributions of pyruvic acid photolysis for that campaign only take account of the acetaldehyde sources that are represented in the model, which are probably incomplete.
We agree that the extrapolation to the HUMPPA-campaign was associated with great uncertainty and have removed this from the paper.

Although the presented work, and the previous study of Eger et al. (2020), provide interesting evidence for an important role for pyruvic acid in the boreal forest environment, the model used in the present modeling study is too simplistic to allow the reported quantitative conclusions to be drawn. The authors should consider addressing the shortcomings and omissions in the chemical mechanism and represented precursor species, possibly by using a customized version of an existing tool such as the MCM. I agree with the recommendations for further experimental studies on the photochemistry of pyruvic acid and on its emission rate, but these recommendations already appear in Eger et al. (2020).
We have now performed a more detailed box-modelling study using the CAABA/MECCA atmospheric chemistry box model with >600 gas-phase species and ~2000 gas-phase reactions and photolysis steps. CAABA/MECCA uses a reduced mechanism based on the MCM.

Specific comments

The chemical mechanism used for the reported simulations (Table S2) contains only selected reactions. Some of these have incomplete product sets and some are parameterized, and I can find no clear discussion or justification of what these are based on or why they are considered adequate. With the exception of pyruvic acid photolysis itself, very little is represented explicitly or fully. As a result, there is a potentially enormous amount of missing organic chemistry which could otherwise contribute to the formation of the species of interest (i.e. acetaldehyde and peroxy radicals, including HO2 and CH3C(O)O2), suggesting that the simulated contributions arising from pyruvic acid photolysis are consistently overestimated. One very clear indication of missing organic chemistry is the CH3C(O)O2 budget reported in sections 3.2 and 3.3. Decomposition of the (observed) PAN is calculated to be the major or dominant CH3C(O)O2 source. However, because PAN is a reservoir (rather than a primary source) this is approximately balanced by CH3C(O)O2 loss by reaction with NO2, indicating that an equivalent primary source is required from elsewhere. This is clearly not fully represented in the simulations.
We now consider the formation of $CH_3C(O)O_2$ in detail and find that the contribution of pyruvic acid to $CH_3C(O)O_2$ production rates are reduced as the referee correctly points out, but are still significant.

As discussed in the literature (e.g. Fischer et al., doi.org/10.5194/acp-14-2679-2014), the sources (of $CH_3C(O)O_2$) can include oxidation or photolysis of co-called "immediate" precursor carbonyls (e.g. acetaldehyde, methylglyoxal, acetone, MEK) and a suite of terpene and isoprene oxidation products. In practice, CH3C(O)O2 can be formed from the reactions of O3 with a-pinene, limonene, 2-carene and 3-carene (i.e. the measured terpenes and also many other BVOCs), from the further chemistry of peroxy radicals containing the C(OO)C(=O)CH3 substructure. These are formed as co-products with OH, following decomposition of relevant Criegee intermediates and reaction of the resultant vinoxy fragment with O2. The relevant chemistry in the applied chemical mechanism, O3 + terpenes —-> OH

therefore omits the C10 organic radical co-product and all its associated organic chemistry. The missing chemistry for this pathway (and for other pathways) not only includes sources of CH3C(O)O2 and other peroxy radicals, but the set of ozonolysis pathways also potentially produces "immediate" CH3C(O)O2 precursors such as methylglyoxal (and larger C(O)C(O)CH3 species) and acetone. In general terms, the OH- O3- and NO3-initiated chemistry represented for monoterpene oxidation is severely limited, inadequately parameterized or completely absent, and the chemistry of other BVOCs (e.g. sesquiterpenes) is also not considered, even though reported to be significant at SMEAR II (Hellén et al., doi.org/10.5194/acp-18-13839-2018).

We now consider the formation of $CH_3C(O)O_2$ in detail, including those pathways associated with terpene degradation insofar as they are implemented in the MCM and CAABA/MECCA.

It is noted that the authors confirm that there must be missing sources of CH3C(O)O2 in their model (final sentence of section 3.3), to account for the observed formation of CH3C(O)OOH during HUMPPA reported by Crowley et al. (doi:10.5194/acp-18-13457- 2018). In view of this, it is not clear why the reported contributions of pyruvic acid photolysis to CH3C(O)O2 formation (e.g. in the Abstract) are not adjusted downwards, or at least qualified, to reflect this. Other than pyruvic acid photolysis, the only sources of acetaldehyde represented in the model appear to be the reactions of OH with ethane, propane and n-butane. These are highly parameterized, only making fractional yields of acetaldehyde and no other products, and incorrectly acting as a radical sink. Other than a brief footnote to Table S2, no justification for this representation is given. There are potentially other precursors to acetaldehyde that may be individually or collectively important, including ethanol, larger oxygenates (e.g. propanal and MEK: Hellén et al., doi.org/10.5194/acp-4-1771- 2004), any species with a C=CHCH3 substructure (e.g. propene, 2-butenes, 2-butenal, 2-hexenal: Hellén et al., doi.org/10.5194/bg-3-167-2006, doi.org/10.5194/acp-4-1771- 2004) and additional alkanes to those already represented.

The degradation chemistry for the alkanes is now treated properly (no longer heavily parameterised) and a much more detailaed reaction scheme (involving oxidised organics such as MEK) has been used.

Other comments
Page 2, line 13: Should probably also include CO2 for completeness. The description in section 2.2.1 also identifies formation of CH3C(O)OH + CO as an "important" channel. If this is the case, should these products also be listed here?
This section has been re-written and the description (based on existing literature) of the possible photolysis products has been moved to a new extended section (1.1)

Page 2, line 21: CH3C(O)O2 and HO2 (i.e. the other pyruvic acid photolysis products of interest) are also more immediate precursors to PAA.
This is correct, but this paragraph deals with the possibility of constraining $CH_3CHO$ mixing ratios by measuring PAA. We now write "This finding was supported by the simultaneous measurement of PAA (which is formed e.g. via the degradation of acetaldehyde in remote environments)"

Page 3, line 2: Is "large" emphasizing that the biogenics are large (i.e. monoterpenes rather than isoprene) or indicating that the emissions are large? It is not clear.
We have amended the text and write "an area that is characterised by large emission rates of biogenics (mainly monoterpenes) and low $NO_x$ concentrations"

Page 3, line 15: Presumably, photolysis rates of other species were also required and were/could be calculated in the same way. The reaction listing in Table S2 also includes photolysis of glyoxal and H2O2 (although the former photolysis rate is based on that of NO2), but somewhat surprisingly not CH3CHO. There are probably other omissions too, such that the species for which photolysis is represented seem rather arbitrary.

Yes, this selection of J-values was exemplary rather than comprehensive and we have removed it. We now write "Photolysis rate coefficients were derived using actinic flux measurements from a spectral radiometer (METCON GmbH)......." The new model considers the photolysis of 325 gas-phase species.

Page 4, lines 17-32: Some of the presented information would seem to require associated citations. Currently, there are none.

This text has been moved to section 1.1 and is more extensive with properly citations to the literature studies.

Page 4, reaction (R5): The products of this reaction should be HO2 + CO2.

Correction made

Page 4, line 31: For clarity and consistency, "CH3CO3 + HO2" should be written "CH3C(O)O2 and HO2".

Correction made

Page 5, line 6: "UPAC".

Correction made (IUPAC)

Page 5, line 9, and Table 1: "CH3CO3" should be "CH3C(O)O2" for consistency. Please also check whole paper for consistency.

Throughout the manuscript all occurrences of $CH_3CO_3$ have been converted to $CH_3C(O)O_2$

Page 5 and Table 1: The considered products of pyruvic acid photolysis are given as either "CH3CHO" or "CH3C(O)O2 + HO2". Although these are the products of interest, they do not describe either the primary photolysis products, or the full set of products following secondary chemistry (which I think are "CH3CHO + CO2" and "CH3C(O)O2 + HO2 + CO2").

We now deal with the products of pyruvic acid in detail in section 1.1 (including the formation of methylhydroxy carbene) and in the supplement (Table S1).

Page 5, lines 1-10. From what is written, it is not clear why the IUPAC recommendation differs so much from the recent study of Reed Harris et al. (2017). Perhaps the studies on which the IUPAC recommendation is based should also be cited and described, as IUPAC presumably judged those to be more reliable and convincing.

The literature on pyruvic acid photolysis is now treated in greater detail in section 1.1 with an additional table in the supplement (S1).

Pages 5 and 6, sections 2.2.2 and 2.2.3. It should probably be stated again clearly what measurements were used to constrain the model for each of the campaign simulations, so that the additions and omissions can be placed in context.

We describe the new box-model in section 2.3 and also list all parameters and trace gases that were used to constrain the model. The mechanism will be made available for download.

Page 5, line 24: "OH" should be "The concentration of OH".
Correction made.

Page 5, lines 28 and 29: The basis for the assigned additional OH reactivity for unmeasured OVOCs sounds rather arbitrary. In practice, there could an abundance of both missing sources and missing sinks of OH that are unaccounted for in the simple model used.
We now calculate OH using the detailed model and state that this compares well with the concentrations calculated from the parameterisation developed for this site using UVB radiation levels. We no longer nudge the losses of OH.

Page 6, line 27: delta-limonene should probably be d-limonene. Is delta-carene 2- carene or 3-carene?
Corrected. We now write (49 % α-pinene, 13 % β-pinene, 27 % carene (sum of 2-carene and 3-carene) , 3 % d-limonene and 8 % camphene)

Page 10, line 2: Should "preceding" be "proceeding"?
Correction made

Page 11, lines 14, 16 and 22. Insert "photolysis" after "pyruvic acid".
Correction made

---

## Author Comment (AC2) · 17 Jun 2021

**Referee 2**

We thank the referee for detailed and helpful comments which are repeated below in black text. Our replies are in blue. We note that both reviewers suggested that making quantitative statements regarding the impact of pyruvic acid photolysis on e.g. $CH_3CHO$ or radical production was not possible given the very concise reaction scheme used. In the light of these comments we have re-done the box-modelling using a comprehensive reaction mechanism drawn from the MCM.

We have also performed quantum chemical calculations to evaluate the fate of methylhydroxy carbene, now believed to be the major product of pyruvic acid photolysis at actinic wavelengths. The manuscript has thus been substantially rewritten.

The authors use a box model to analyze data from two field studies in Hyytiala (summer-autumn of different years), examining the potential impact of pyruvic acid (PYR) photolysis on acetaldehyde and HOx radical budgets in this boreal forest area. The topic is of interest and importance, and thus relevant to ACP readership. I have some significant concerns with the work, however, mostly related to assumptions made in the box model analyses; some of these are discussed by the authors, but I think there are others that need to be addressed (particularly in the context of how quantitative the conclusions are, given the uncertainties in PYR photolysis and depositional loss and the nature of some of the approximations made). More details follow.

We have now performed a more detailed box-modelling study using the CAABA/MECCA atmospheric chemistry box model with >600 gas-phase species and ~2000 gas-phase reactions and photolysis steps. CAABA/MECCA uses a reduced mechanism based on the MCM.

One over-arching comment:

Some of the ideas explored here have already been touched upon in the authors' previous publication on this topic (Eger et al., ACP 2020), and I found that some of the questions that came to mind were actually addressed in the previous paper. I would thus like to see a clear distinction made between the two works, via the addition of a paragraph or two to the end of the introductory material to summarize the findings of the previous study, and to set the stage for what is done in this paper.

The paper has been substantially revised. Along with the detailed box-modelling we have now done, we have also performed a theoretical analysis of the fate of methylhydroxy carbene formed initially in pyruvic acid photolysis. Overlap between the papers is substantially reduced.

More detailed comments:

Is it realistic to assume a mixed boundary layer (even in daytime) for fairly short-lived vegetative emissions? Can the authors be more quantitative or descriptive of what the implications of this assumption are?

As we describe in detail, we have attempted to use realistic boundary-layer heights as this in central to the calculation of deposition velocities and emission rates for pyruvic acid and monoterpenes. If the boundary layer is not well mixed, the emission rates we derived (based on measurements close to ground level) will be biased high and the deposition losses biased

low. We now write "Further, our calculations are based on the assumption that the sources for pyruvic acid and MT emissions are evenly distributed and measurements made at ~ 8.5 m above the ground are representative of the entire boundary layer (i.e. that the boundary layer is well-mixed, including the very shallow boundary layer at night). A gradient in pyruvic acid mixing ratios at night cannot be ruled out, which would impact on our results. If the boundary layer is not well mixed, the emission rates we derived will be biased high and the deposition losses biased low. We estimate that the emission ratio ($E_{pyr}$ / $E_{MT}$) in Eq. (2) is associated with an overall uncertainty of a factor ~2.

Is the chemistry in the box model sufficiently detailed to capture HOx radical budgets accurately in this complex terpene-driven system, and thus assess changes driven by PYR photolysis? In particular, could there not be multiple sources of CH3C(O)O2 radicals from the terpene chemistry that is not included in the model? Do we really know enough about terpene oxidation to rule out PYR production?

We have now performed a more detailed box-modelling study using the CAABA/MECCA atmospheric chemistry box model with >600 gas-phase species and ~2000 gas-phase reactions and photolysis steps. We now consider the formation of $CH_3C(O)O_2$ in detail, including those pathways associated with terpene degradation insofar as they are implemented in the MCM and CAABA/MECCA.

Regarding OH in the IBAIRN modeling, I did not quite follow sentences near the bottom of page 5. Am I correct that OH is modeled (not constrained), but is lowered in the model via addition of extra reactivity to match the UV/OH correlation determined in other campaigns at the site? Please clarify.

It is correct that OH was not measured during IBAIRN. However, many years of OH measurements at this site have been used to derive a simple expression that reproduces OH levels reasonably well from measurements of UVB. Our new box-model runs generate OH levels that are in agreement with those based on the UVB calculation and this is now stated in the text.

Bottom of page 6 - Do the monoterpenes (MT's) have significantly different lifetimes and, if so, can the anything be said about how averaging their loss rates might impact the analysis?

We no longer quote the average lifetime as this information is superfluous. We recognise that treating the MT non-explicitly introduces significant uncertainty into the calculation and now mention this in the text.

Page 7 - I would like to get some sense of the variability in the average diel profiles shown, particularly for the emission rates of PYR and MT (Figure 2). This variability then carries forward to the data shown in Figures S3 and S4, correct?

This is correct. We now refer to our previous paper (Eger et al, 2020) in which we plot the diel profiles of monoterpenes and pyruvic acid along with the 25 % and 75 % percentiles as an indicator of variability. To maintain clarity of presentation, we have chosen not to include this variability in the present Figure 2.

Bottom of page 7 - The data showing the T-dependence of the PYR to MT emissions ratio look quite convincing. Isn't it likely, however, that this ratio is influenced also by light levels, thus complicating the analysis of the T-dependence?

The temperature dependence is highly uncertain as it is strongly influenced by the deposition term for pyruvic acid, which is dependent on the boundary-layer (and its insolation / temperature dependent variability). In order to emphasise this we now write: "We note that

(like the values of $E_{pyr}$) the temperature dependence derived is strongly influenced by the insolation-dependent diel variation of the MXL height and thus carries significant uncertainty."

Page 9 - Is PAN nominally in steady-state (or close to it)? If so, I am not sure it makes sense to consider its decomposition to be a source of CH3C(O)O2 radicals?
This is correct. In evaluating our new model results, we do not consider PAN as a net source of $CH_3C(O)O_2$

Page 10/11 - I have significant misgivings about doing anything quantitative with the HUMPPA campaign data, given the absence of PYR mixing ratio data. Surely, and as stated (to an extent) by the authors, the emissions ratios (E_PYR / E_MT) could be affected by changes in temperatures, light levels, vegetation, availability of plant litter, different soil moisture content, etc. etc. I would recommend nothing more than some qualitative estimates regarding the HUMPPA campaign data, rather than any sort of quantitative analysis. Also, I realize that inclusion/exclusion of the biomass burning impacted data made little difference to the findings, but is it not quite likely that there are biomass burning sources of acetaldehyde, MTs and possible PYR during these periods that are not factored into the analysis here?
We agree that the use of pyruvic emissions derived in the autumn to the same site in the summer carries great uncertainty, and we have removed the HUMPA analysis from the paper.

Minor comments:

Page 4 - (R5) should have CO2 as the product, not CO.
Correction applied

Page 5, line 6 – IUPAC is missing the I.
Correction applied

Page 6, line 14 – Can some data be shown to demonstrate the statement made here stating that changes in the PYR mixing ratio could be reproduced by the model?
In our new box-model runs pyruvic acid was always constrained by observations. We no longer make this statement.

Pg. 7, line 9 - There is roughly a factor of two uncertainty in PYR deposition, and there is roughly a factor of four difference in the overall PYR quantum yield used in the different sensitivity studies. Thus, would not the uncertainty in the emission ratio be larger than a factor of two?
The uncertainty in the relative emission ratio depends on the contribution to losses by photolysis and by deposition. Note that uncertainty in the BL-height cancel with this approach. We have not considered the different quantum yields. We have adopted the text to clarify this.

Page 8, line 25 – I am getting different numbers for the emissions ratios than are given here (on the basis of the branch emission rates given). Please check / confirm.
The emission-ratios quoted are based on the emission rates (in µg C $gdw^{-1}$ $h^{-1}$) listed in Table 2 and take into account the number of carbon atoms in pyruvic acid (3), isoprene (5) and monoterpenes (10).

Page 9, line 29 – I think you are meaning to say that the fractional contribution of the alkanes drastically decreases?

Based on the new box-model results, this text has been re-written.

Lastly, a bit of an aside: The authors might not yet be aware, but there was a presentation at the AGU a couple of months ago that appears to show the hydroxycarbene CH3COH + CO2 as the major channel at 351 nm (Osborne and co-workers). I don't think any quantum yields were reported.

We thank the reviewer for this information. The work was recently published (Samanta et al, 2021, Phys. Chem. Chem. Phys.). As this result changes the picture substantially, we have performed quantum chemical calculations on the fate of the carbene and changed box-model scenario "C" to consider carbene chemistry. This is a totally new section in the paper.

---

## Referee Report (RR1)

Review of acp-2020-975 revision: "Impact of pyruvic acid photolysis on acetaldehyde and peroxy radical formation in the boreal forest: Theoretical calculations and model results." by Eger et al..

**General comments**

This is a revised and updated paper, presenting an analysis of the impact of pyruvic acid photolysis on the chemistry and composition of the boreal forest boundary layer, for the conditions of a 2016 autumn campaign (IBAIRN) at the SMEAR II field station in Hyytiälä, Finland. The observationally-constrained box modelling analysis has been substantially improved by use of a detailed and previously peer-reviewed chemical mechanism (Sander et al., 2019), allowing a much more reliable and informative assessment of the contributions of pyruvic acid photolysis to the formation of acetaldehyde, peroxy radicals (HO2 and CH3C(O)O2) and related products at the campaign location. The work has also benefitted greatly from being able to take account of the recent experimental study of Samanta et al. characterising the primary formation of methylhydroxy (2021), carbene (CH3COH) and CO2 from the photodissociation of pyruvic acid, a piece of work presented at the AGU and published since my previous review in early December 2020. The results of that study are further enhanced by the inclusion of a theoretical assessment of the fate of CH3COH under tropospheric boundary layer conditions. This is now a very complete piece of work, providing a thorough analysis of the impact of pyruvic acid photolysis on the chemistry and composition of the boreal forest boundary layer during the IBAIRN campaign. This paper is entirely suitable for publication in ACP.

Some comments on the revised paper are given below. These are mainly minor and typographical, but with one or two suggestions of where some further information might be helpful to the inquisitive reader:

line 94: "unumolecular".

line 145: "...monoterpenes (henceforth referred to as MT)...". A very minor point, but I note that the term "monoterpenes" is used a further 9 times, although the abbreviation "MT" is subsequently used 17 times.

Line 185: I am reassured that the chemical mechanism is based on that for CAABA/MECCA, as previously reported in Sander et al. (2019), and therefore a vast improvement on that used in the original analysis reported in the first version of this paper. In view of the importance of monoterpenes in this work ( $\alpha$ -pinene,  $\beta$ pinene, 2- + 3-carene, limonene and camphene are reported as detected on line 228), some information should be given on how the speciation was constrained in the model. I note that the mechanism in Sander et al. (2019) includes explicit  $\alpha$ - and  $\beta$ -pinene chemistry (partly informed by MCM), and chemistry for carene and camphene that feeds into the pinene mechanisms. There does not appear to be any limonene chemistry, although that is in MCM. I'm sure these points are clarified in the "complete reaction scheme and source of rate coefficients" in the data archive for this paper, but that appears to be unavailable without contacting the authors (which is incompatible with anonymous review). Some brief information in the main text/SI about monoterpene speciation and chemistry in the model calculations would be helpful.

Line 195: I agree that peroxyacyl nitrate burdens are invariably dominated by PAN itself, but a 90 % contribution seems a little high to me and requires more justification. PAN/(total PANs) increases with processing time because many larger PANs (e.g., PPN, MPAN) and their precursors are degraded to species that form PAN. Therefore, I'm not sure ratios based on airborne measurements over the Arctic (Roiger et al., 2011) and the Pacific (Roberts et al., 2004) are necessarily an appropriate quide. Based on Williams et al. https://doi.org/10.1029/97GL00548, 1997) and Roberts et al. (doi:10.1029/2001JD000947, 2002), PAN/PPN seems to be around 5 or 6 in relatively young anthropogenic dominated air masses, consistent with an upper limit contribution of a bit less than 90 % (upper limit because there are higher PANs too); and in biogenic (isoprene) dominated environments, the PAN/MPAN ratio is typically 4-10 (again probably depending on processing time). The average contributions over all conditions in the Roberts et al. (2002) SOS study are about 80% PAN, 11% PPN, 2% PiBN and 7% MPAN. Detection of other higher PANs has also been reported (e.g., PBN by Grosjean et al., https://doi.org/10.1021/es00039a013, 1993), and the oxidation of the monoterpenes is expected to make large PANs (e.g., those derived from pinonaldehyde, limononaldehyde and caronaldehyde). In view of this, how sensitive are the calculations to (e.g.) a 10 % change in the assigned PAN contribution? In addition, the model should allow a speciation to be calculated. I count about 35 PANs in the mechanism in Sander et al. (2019). I realise that many will be unimportant for the IBAIRN conditions, but they include PAN, PPN, PiPN, MPAN, the oxygenated species,  $HOCH_2C(0)OONO_2$ ,  $HC(0)CH_2C(0)OONO_2$ , small and those derived from pinonaldehyde and norpinonaldehyde. The modelled speciation is something that could be informative in a wider context, and which could be reported and applied in this work.

line 229: Should " $\Delta$ -" be "d-" (or "D-") for limonene?

line 230: "sifnificant".

Line 326: "coeffocient".

lines 328-329: "...is reminiscent of the chemistry of other  $\alpha$ -OH alkyl radicals...". I understand the mechanistic point, but can CH3C•(OH)OO• be described simply as an  $\alpha$ -OH alkyl radical? It is a biradical, which looks like a Criegee intermediate

(biradical/zwitterion). If so, I think it would decompose/rearrange to form either a dioxirane (anti-) or PAA (syn-) (based on Table 28 in the SI of Vereecken et al., https://doi.org/10.1039/c7cp05541b, 2017). The products presented here (formation of  $CH_3CO + HO_2$ ) are compatible with formation and fragmentation of hot PAA, so is this effectively the same species and process reported by Vereecken et al. (2017) and is  $CH_3C \cdot (OH)OO \cdot$  a Criegee? If so, some clarification of this, and reference to Vereecken et al. (2017), might be helpful. Regarding the phrase on line 328-329 (in quotes above), perhaps omitting the word "other" would help.

line 380: A reference would seem to be required for organic acid concentrations. What about Millet et al. (https://doi.org/10.5194/acp-15-6283-2015)?

line 427: "neglecetd".

line 428: ". (see Fig. S"

line 433: Is the word "coincidentally" necessary here? It is logical that the NO/NOx ratio tends to maximise in the middle of the day when  $NO_2$  photolysis is most rapid.

line 449: "productio".

line 456: "indrect".

line 463: For consistency, and equation balancing, the HCHO photolysis reaction needs to specify two  $O_2$  molecules in a bracket.

line 470: I think "and HCHO" needs to be deleted here, because there are too few values given and HCHO is given a value in the next sentence.

Line 471: "enhanved".

Lines 475-478: The first sentence of the conclusions could be much clearer. A comma after "major product of its photodissociation" would make it clearer, but I suggest splitting the sentence up into two or three sentences might be helpful.

Line 481: This information should probably be qualified to reflect that the  $CH_3COH$  +  $O_2$  rate coefficient was elevated by an order of magnitude over the calculated value to make it contribute.

---

## Author Response (AR2)

**Referee 1**

The referee's comment are in black, our replies in blue and changes to the text in red.

General comments This is a revised and updated paper, presenting an analysis of the impact of pyruvic acid photolysis on the chemistry and composition of the boreal forest boundary layer, for the conditions of a 2016 autumn campaign (IBAIRN) at the SMEAR II field station in Hyytiälä, Finland. The observationally-constrained box modelling analysis has been substantially improved by use of a detailed and previously peer-reviewed chemical mechanism (Sander et al., 2019), allowing a much more reliable and informative assessment of the contributions of pyruvic acid photolysis to the formation of acetaldehyde, peroxy radicals (HO2 and CH3C(O)O2) and related products at the campaign location. The work has also benefitted greatly from being able to take account of the recent experimental study of Samanta et al. (2021), characterising the primary formation of methylhydroxy carbene (CH3COH) and CO2 from the photodissociation of pyruvic acid, a piece of work presented at the AGU and published since my previous review in early December 2020. The results of that study are further enhanced by the inclusion of a theoretical assessment of the fate of CH3COH under tropospheric boundary layer conditions. This is now a very complete piece of work, providing a thorough analysis of the impact of pyruvic acid photolysis on the chemistry and composition of the boreal forest boundary layer during the IBAIRN campaign. This paper is entirely suitable for publication in ACP. Some comments on the revised paper are given below. These are mainly minor and typographical, but with one or two suggestions of where some further information might be helpful to the inquisitive reader:

We thank the referee for this very positive assessment of the revised manuscript.

line 94: "unumolecular".

Corrected

line 145: "...monoterpenes (henceforth referred to as MT)...". A very minor point, but I note that the term "monoterpenes" is used a further 9 times, although the abbreviation "MT" is subsequently used 17 times.

We have now used the abbreviation MT or MTs throughout

Line 185: I am reassured that the chemical mechanism is based on that for CAABA/MECCA, as previously reported in Sander et al. (2019), and therefore a vast improvement on that used in the original analysis reported in the first version of this paper. In view of the importance of monoterpenes in this work (α-pinene, βpinene, 2- + 3-carene, limonene and camphene are reported as detected on line 228), some information should be given on how the speciation was constrained in the model. I note that the mechanism in Sander et al. (2019) includes explicit α- and β-pinene chemistry (partly informed by MCM), and chemistry for carene and camphene that feeds into the pinene mechanisms. There does not appear to be any limonene chemistry, although that is in MCM. I'm sure these points are clarified in the "complete reaction scheme and source of rate coefficients" in the data archive for this paper, but that appears to be unavailable without contacting the authors (which is incompatible with anonymous review). Some brief information in the main text/SI about monoterpene speciation and chemistry in the model calculations would be helpful.

The MTs were split according to the GC-AED measurements. We have added the following text:

Based on the GS-AED measurements, the MTs were split into α-pinene (49 %), β-pinene (13 %), Δ-carene (27 %) and camphene (8 %). Limonene is not included in the standard chemical mechanism of CAABA/MECCA but as its contribution to the MTs was only 3 % it was treated as Δ-carene (increasing its contribution to 30 %).

Line 195: I agree that peroxyacyl nitrate burdens are invariably dominated by PAN itself, but a 90 % contribution seems a little high to me and requires more justification. PAN/(total PANs) increases with processing time because many larger PANs (e.g., PPN, MPAN) and their precursors are degraded to species that form PAN. Therefore, I'm not sure ratios based on airborne measurements over the Arctic (Roiger et al., 2011) and the Pacific (Roberts et al., 2004) are necessarily an appropriate guide. Based on Williams et al. https://doi.org/10.1029/97GL00548, 1997) and Roberts et al. (doi:10.1029/2001JD000947, 2002), PAN/PPN seems to be around 5 or 6 in relatively young anthropogenic dominated air masses, consistent with an upper limit contribution of a bit less than 90 % (upper limit because there are higher PANs too); and in biogenic (isoprene) dominated environments, the PAN/MPAN ratio is typically 4-10 (again probably depending on processing time). The average contributions over all conditions in the Roberts et al. (2002) SOS study are about 80% PAN, 11% PPN, 2% PiBN and 7% MPAN. Detection of other higher PANs has also been reported (e.g., PBN by Grosjean et al., https://doi.org/10.1021/es00039a013, 1993), and the oxidation of the monoterpenes is expected to make large PANs (e.g., those derived from pinonaldehyde, limononaldehyde and caronaldehyde). In view of this, how sensitive are the calculations to (e.g.) a 10 % change in the assigned PAN contribution?

The referee is correct that 90% may be on the high side. Unfortunately, there are no measurements of the fractional contribution of PAN to PANs at Hyytiälä and use of data from an isoprene dominated environment (with high MPAN) is unlikely to be representative. In any case, PAN, which is in equilibrium with acetylperoxy radical and $NO_2$, does not contribute to the net $CH_3CO$ production rate (see Figure 7) and changing its mixing ratio by a few percent will not impact our results or conclusions.

In addition, the model should allow a speciation to be calculated. I count about 35 PANs in the mechanism in Sander et al. (2019). I realise that many will be unimportant for the IBAIRN conditions, but they include PAN, PPN, PiPN, MPAN, the small oxygenated species, HOCH2C(O)OONO2, HC(O)CH2C(O)OONO2, and those derived from pinonaldehyde and norpinonaldehyde. The modelled speciation is something that could be informative in a wider context, and which could be reported and applied in this work.

We agree that the speciated, modelled PANs would be useful for comparison with speciated measurements of PANs. However, no such measurements were available during IBAIRN, and we feel that this extra information would detract from the focus of this paper, which is on $CH_3CHO$, $CH_3C(O)O_2$ and $HO_2$ formation from PA photolysis.

line 229: Should "Δ-" be "d-" (or "D-") for limonene?
Correction made

line 230: "sifnificant".
Correction made

Line 326: "coeffocient".
Correction made

lines 328-329: "...is reminiscent of the chemistry of other α-OH alkyl radicals...". I understand the mechanistic point, but can CH3C•(OH)OO• be described simply as an α-OH alkyl radical? It is a biradical, which looks like a Criegee intermediate (biradical/zwitterion). If so, I think it would decompose/rearrange to form either a dioxirane (anti-) or PAA (syn-) (based on Table 28 in the SI of Vereecken et al., https://doi.org/10.1039/c7cp05541b, 2017). The products presented here (formation of CH3CO + HO2) are compatible with formation and fragmentation of hot PAA, so is this effectively the same species and process reported by Vereecken et al. (2017) and is CH3C•(OH)OO• a Criegee? If so, some clarification of this, and reference to Vereecken et al. (2017), might be helpful.

The specific chemistry for a Criegee intermediate is due to its singlet wavefunction (best described as CH3C(OH)=O+O-), which allows for re-arrangement with new bond formation in the dioxirane or PPA products mentioned. The CH3C•(OH)OO• intermediate described here, however, must instead have a triplet wavefunction as the reaction of CH3COH + 3O2 occurs on the triplet surface. The same-spin electrons prevent formation of the needed new bonds after rearrangements and because of this, unimolecular reactions are energetically not accessible and the fate of the triplet intermediate is reaction with O2. We now note explicitly in the paper that the triplet and singlet forms have very distinct chemistries.

The decomposition of the $CH_3C^{\bullet}(OH)OO^{\bullet}$ triplet diradical intermediate, forming $CH_3C^{\bullet}=O$ + $HO_2$, is reminiscent of the chemistry of α-OH alkyl radicals with unpaired electrons, and should occur rapidly owing to the sufficiently high energy content of the peroxyl-alkyl diradical (Hermans et al., 2005, 2004; Dillon et al., 2012; Olivella et al., 2001; Dibble, 2002). Note that this chemistry is very distinct from that of the singlet $CH_3C(OH)OO$ Criegee intermediate.

Regarding the phrase on line 328-329 (in quotes above), perhaps omitting the word "other" would help.
"Other" has been deleted.

line 380: A reference would seem to be required for organic acid concentrations. What about Millet et al. (https://doi.org/10.5194/acp-15-6283-2015)?
Reference to Millet et al, 2015 added

line 427: "neglecetd".
Correction made

line 428: ". (see Fig. S" line 433: Is the word "coincidentally" necessary here? It is logical that the NO/NOx ratio tends to maximise in the middle of the day when NO2 photolysis is most rapid.
"coincidentally" has been deleted.

line 449: "productio".
Correction made

line 456: "indrect".
Correction made

line 463: For consistency, and equation balancing, the HCHO photolysis reaction needs to specify two O2 molecules in a bracket.
Correction made

line 470: I think "and HCHO" needs to be deleted here, because there are too few values given and HCHO is given a value in the next sentence.

Correction made

Line 471: "enhanved".

Correction made

Lines 475-478: The first sentence of the conclusions could be much clearer. A comma after "major product of its photodissociation" would make it clearer, but I suggest splitting the sentence up into two or three sentences might be helpful.

We have modified the sentence and now write:

We have combined measurements of pyruvic acid in an autumn campaign in the boreal forest (IBAIRN) with theoretical calculations designed to characterise the fate of the methylhydroxy carbene radical ($CH_3COH$, the major product of its photodissociation) with a box modelling study. We investigated the impact of pyruvic acid photolysis on the rates of production of acetaldehyde ($CH_3CHO$) and the peroxy radicals $CH_3C(O)O_2$ and $HO_2$.

Line 481: This information should probably be qualified to reflect that the CH3COH + O2 rate coefficient was elevated by an order of magnitude over the calculated value to make it contribute.

We have modified the text to write:

The reaction of $CH_3COH$ with $O_2$ is slow, but will contribute to its fate (and thus the formation of $CH_3C(O)O_2$ and $HO_2$) in the lower atmosphere where $O_2$ concentrations are high if the rate constant used (elevated by an order of magnitude compared to the highly uncertain theoretical value) is correct.

**Referee 2**

The referee's comment are in black, our replies in blue and changes to the text in red.

The manuscript is vastly improved; the authors have done a great job of considering (via theoretical methods and box modeling) the chemistry of CH3C:OH which has recently been reported to be a major pyruvic acid photolysis product. I think the manuscript is acceptable essentially as is, and I have only a few minor corrections to suggest:
We thank the referee for this very positive assessment of the revised manuscript.

Line 245: Should be J-Pyr, not J-NO2?
Corrected.

Line 347: Maybe mention that this is a formate ester.
We have added the chemical formula to the text.
....forming a 1-hydroxyethylester (CH3CH(OH)OC(O)H).

Page 13, It might be helpful to indicate that the text from line 392 to the end of the section all belongs to Scenario B (i.e., involves the CH3COH chemistry).
The text referred to is preceeded by "Scenario B" . We have removed a paragraph-break and the text "in the box-model" so that it is clear that the rest of the section deals with Scenario B.
In scenario B, we consider the effects of using photodissociation quantum yields of 0.2, 0.5 and 1 (scenarios $B_{0.2}$, $B_{0.5}$ and $B_1$, respectively). Photolysis at wavelengths < 340 nm was considered...etc.

Also, for scenario A, repeating the IUPAC quantum yield information would be helpful, I think.
We have added this information as suggested.
**Scenario A**: In this scenario we used pyruvic acid cross sections, quantum yields and product yields according to the IUPAC recommendations (IUPAC, 2020) with a photodissociation quantum yield ($\phi$) of 0.2 at 1 bar pressure and branching ratios of 0.6, 0.05 and 0.35 for reactions R1, R2 and R3  as listed in section 1.1.

Line 470: Delete 'and HCHO' ?
Correction made

Also, I think these data are in Fig S6, not S5.
Correction made

There are typos on lines 94, 131, 327, 428, 449, 456, 471, 484.
Typos found and corrected